# Developing and optimizing shrub parameters representing sagebrush (*Artemisia* spp.) ecosystems in the Northern Great Basin using the Ecosystem Demography (EDv2.2) model

Karun Pandit[1], Hamid Dashti[1], Nancy F. Glenn[1], Alejandro N. Flores[1], Kaitlin C. Maguire[2], Douglas J. Shinneman[2], Gerald N. Flerchinger[3], Aaron W. Fellows[3]

[1]Department of Geosciences, Boise State University, 1910 University Dr, Boise, ID 83725-1535 USA
[2]United States Geological Survey, Forest and Rangeland Ecosystem Science Center, 970 Lusk St., Boise, ID 83706
[3]United States Department of Agriculture, Agricultural Research Service, 800 Park Blvd., Suite 105, Boise, ID 83712

*Correspondence to*: Karun Pandit (karunpandit@gmail.com)

**Abstract.** Ecosystem dynamic models are useful for understanding ecosystem characteristics over time and space because of their efficiency over direct field measurements and applicability to broad spatial extents. Their application, however, is challenging due to internal model uncertainties and complexities arising from distinct qualities of the ecosystems being analyzed. The sagebrush-steppe in western North America, for example, has substantial spatial and temporal heterogeneity as well as variability due to anthropogenic disturbance, invasive species, climate change, and altered fire regimes, which collectively make modelling dynamic ecosystem processes difficult. Ecosystem Demography (EDv2.2) is a robust ecosystem dynamic model, initially developed for tropical forests, that simulates energy, water, and carbon fluxes at fine scales. Although EDv2.2 has since been tested on different ecosystems via development of different Plant Function Types (PFT), it still lacks a shrub PFT. In this study, we developed and parameterized a shrub PFT representative of sagebrush (*Artemisia* spp.) ecosystems in order to initialize and test it within EDv2.2, and to promote future broad-scale analysis of restoration activities, climate change, and fire regimes in the sagebrush-steppe. Specifically, we parameterized the sagebrush PFT within EDv2.2 to estimate gross primary production (GPP), using data from two sagebrush study sites in the northern Great Basin. To accomplish this, we employed a three-tier approach: 1) To initially parameterize the sagebrush PFT, we fitted allometric relationships for sagebrush using field-collected data, information from existing sagebrush literature, and parameters from other land models. 2) To determine influential parameters in GPP prediction, we used a sensitivity analysis to identify the five most sensitive parameters. 3) To improve model performance and validate results, we optimized these five parameters using an exhaustive search method to estimate GPP, and compared results with observations from two Eddy Covariance (EC) sites in the study area. Our modelled results were encouraging, with reasonable fidelity to observed values, although some negative biases (i.e., seasonal underestimates of GPP) were apparent. Our finding on preliminary parameterization of the sagebrush shrub PFT is an important step towards subsequent studies on shrubland ecosystems using EDv2.2 or any other process-based ecosystem models.

## 1 Introduction

Ecosystem dynamic models have been widely used to estimate terrestrial carbon flux and to project ecosystem characteristics over time and space (Dietze et al., 2014; Fisher et al., 2018), largely due to their efficiency over direct field measurements and their applicability to broad spatial scales. However, these models have also been associated with high levels of internal uncertainty and questions regarding their applicability to distinct and often complex ecosystems at large scale (Kwon et al., 2008). Sagebrush (*Artemisia* spp.) ecosystems in Western North America provide a good example of these types of modelling challenges, as these ecosystems are spatially heterogeneous and shaped by complex dynamics over time. Sagebrush ecosystems hold both high ecological and socio-economic value, but they have been reduced to nearly half of their historical range and are declining at an alarming rate (Knick et al., 2003; Schroeder et al., 2004). Various factors have contributed to this decline, including land clearing, invasion of nonnative species such as cheatgrass (*Bromus tectorum*), and climate change, that have collectively altered vegetation composition, hydrological function, and wildfire frequency (Bradley, 2010; Connelly et al., 2004; McArthur and Plummer, 1978; Schlaepfer et al., 2014). In an attempt to restore portions of the sagebrush ecosystem, land managers have focused on reducing flammable vegetation, controlling invasive species, and seeding native plant species (Chambers et al., 2014; McIver and Brunson, 2014). There are relatively few studies that have evaluated carbon flux in sagebrush ecosystems in response to prescribed fire or restoration activities, and most of them used observational data from Eddy Covariance (EC) stations. However, given the large spatial extent of the sagebrush biome (>500,000 km$^2$; Miller et al. 2011) and the paucity of EC station sites in sagebrush landscapes, the function of this ecosystem remains poorly understood, especially as management activities, fire, climate change, and invasive species continue to alter ecosystem structure, composition, and spatiotemporal dynamics.

Ecosystem Demography (EDv2.2), is a process-based ecosystem dynamic model that approximates the behaviour of ensembles of size and age-structured individual plants to capture sub-grid level ecosystem heterogeneity using partial differential equations (Medvigy et al., 2009; Moorcroft et al., 2001). This model was originally developed to study tropical ecosystems with trees as a primary component, but it has since been modified and applied to several different ecosystems, including boreal forests (Trugman et al., 2016), and temperate forests (Antonarakis et al., 2014; Medvigy et al., 2009; Medvigy et al., 2013). However, its application to semi-arid shrubland ecosystems has not been explored and it lacks a shrub Plant Function Type (PFT) to study these ecosystems. Thus, we developed and parameterized a sagebrush PFT for EDv2.2, and used it to estimate gross primary production (GPP) for the sagebrush ecosystems in the Reynolds Creek Experimental Watershed (RCEW) located in the Northern Great Basin of the United States, a cold-desert region dominated by expansive, shrub-steppe ecosystems.

In this study, our primary objective was to develop preliminary sagebrush PFT parameters in EDv2.2 and to constrain uncertainties through optimization of selected PFT parameters. To accomplish this, we employed a three-tiered approach. First, we parameterized the sagebrush PFT, by fitting allometric relationships for sagebrush using field-collected data, information from existing sagebrush literature, and borrowing parameters from other land models. Second, to identify the most

influential parameters in GPP prediction, we used a sensitivity analysis and identified the five most-sensitive parameters affecting changes in GPP estimates. Third, to improve upon and assess model performance, we optimized the five most sensitive parameters using an exhaustive search method to estimate GPP, and then compared the results with observations from two Eddy Covariance (EC) sites in the study areas. Our preliminary parameterization of the sagebrush shrub PFT is an important first step towards further study of shrubland ecosystem function using EDv2.2 or similar process-based ecosystem models.

## 2 Material and methods

### 2.1 Ecosystem Demography (EDv2.2) model

EDv2.2 is a process-based terrestrial biosphere model that occupies a mid-point on the continuum of individual-based (or gap) to area-based (or big-leaf) models (Fisher, 2010; Smith et al., 2001). Area-based models like LPJ-DGVM (Lund-Potsman-Jena Dynamic Vegetation Model) (Sitch et al., 2003), and BIOME BGC (Running and Hunt, 1993 as cited in Bond-Lamberty et al., 2014) represent plant communities with area-averaged representation of a PFT for each grid cell. The simplification and computational efficiency of these models make them widely applicable for regional ecosystem analysis; however, this advantage often comes with a limited ability to properly capture light competition and competitive exclusion (Fisher, 2010; Bond-Lamberty et al., 2014; Smith et al, 2001). In contrast, individual-based models, (IBMs) such as JABOWA (Botkin et al., 1972) and SORTIE (Pacala et al., 1993), represent vegetation at the individual plant level, thus making it possible to incorporate community processes like growth, mortality, recruitment, and disturbances. Recent improvements in computational efficiency have permitted the use of IBMs beyond traditional applications confined to limited spatial and temporal scales. EDv2.2 is a cohort based model where individual plants with similar properties, in terms of size, age, and function, are grouped together to reduce the computational cost while retaining most of the dynamics of IBMs. Each cohort is defined by a PFT, number of plants per unit area, and dimensions of a single representative plant like diameter, height, structural biomass, and live biomass (Fisher et al., 2010). The cohort based modelling approach in EDv2.2 has been applied to capture detailed ecological processes in studies investigating the effects of fire, drought, insect infestations, and climate effects on ecosystems at broad spatial scales (Fisher et al, 2018).

The land surface in EDv2.2 is composed of a series of gridded cells, which experience meteorological forcing from corresponding gridded data or from a coupled atmospheric model (Medvigy, 2006). The mechanistic scaling from individual to the region is achieved through size and age structured partial differential equations that closely approximate mean behaviour of a stochastic gap model (Medvigy et al., 2009; Moorcroft et al., 2001). Each grid cell is subdivided into a series of dynamic horizontal tiles, which represent locations that experience similar disturbance history and have an explicit vertical canopy structure. This mechanism helps capture both vertical and horizontal distributions of vegetation structure and compositional heterogeneity compared to area-based models (Kim et al., 2012; Moorcroft et al., 2001; Moorcroft et al., 2003; Sellers et al., 1992). EDv2.2 consists of multiple sub-models for plant growth and mortality, phenology, disturbance, biodiversity,

hydrology, land surface biophysics, and soil biogeochemistry, to predict short-term and long-term ecosystem flux and to represent natural and anthropogenic disturbances (Kim et al., 2012; Medvigy et al., 2009; Zhang et al., 2015). Sub-models in EDv2.2 rely mostly on many PFT-specific parameters, representing unique attributes of that particular group of species, to define the stated biological processes (Knox et al., 2015). Studies on shrub parameterization have been performed in LPJ-GUESS for the tundra region (Miller and Smith, 2012; Wolf, 2008), however, parameterization for shrub PFT is lacking for semi-arid shrubland ecosystems. EDv2.2 has parameters defined for 17 different PFTs including grasses (C3 & C4), conifers, deciduous trees (temperate & tropical), and agricultural crops. In this study, we identified parameters for the sagebrush (shrub) ecosystem to simulate it in the model as a new PFT. We limited the scope of this study to sagebrush PFT parameterization using model structures and processes adopted in EDv2.2 for trees (e.g. seed dispersal, competition, mortality, and phenology), which we assumed would be generally applicable to shrubs (Wolf et al., 2008). Because we explored model performance based on GPP estimates, we selected eleven different parameters related to plant ecophysiology and biomass allocation to conduct sensitivity and optimization assessments (Table 1). We mainly relied on similar studies (Dietze et al., 2014; Fisher et al., 2010; LeBauer et al., 2013; Medvigy et al., 2009; Mo et al., 2008; Pereira et al., 2017), our preliminary sensitivity analyses, and consultation with other developers and users of the EDv2.2 model to select the parameters.

**Table 1**. Parameters used to explore model performance for sagebrush PFT.

| Parameter | Description | Unit |
|---|---|---|
| Maximum carboxylation rate ($V_{m0}$) | Maximum carboxylation rate at 15ºC | $\mu mol\,m^{-2}s^{-1}$ |
| Stomatal slope ($M$) | Slope of stomatal conductance-photosynthesis relationship | - |
| Cuticular conductance ($b$) | Intercept of stomatal conductance-photosynthesis relationship | $\mu mol\,m^{-2}s^{-1}$ |
| Water conductance ($K_w$) | Supply coefficient for plant water uptake | $ms^{-1}kgCroot^{-1}$ |
| Leaf width ($W_{leaf}$) | Controls leaf boundary layer conductance (m) | m |
| SLA | Specific leaf area | $m^2kg^{-1}$ |
| GRF ($r_g$) | Growth respiration factor | - |
| Q-ratio ($q$) | Ratio of fine roots to leaves | - |
| Leaf turnover rate ($\alpha_{leaf}$) | Inverse of leaf life span | Per annum ($a^{-1}$) |
| Fine root turnover rate ($\alpha_{root}$) | Inverse of fine root life span | Per annum ($a^{-1}$) |
| Storage turnover rate ($\alpha_{storage}$) | Turnover rate of plant storage pools | Per annum ($a^{-1}$) |

Detailed descriptions of sub-models of EDv2.2 are available in existing literature (Medvigy et al., 2009; Moorcroft et al., 2001); thus, here we describe the ones related to the parameters used in this study. The ecophysiological sub-model has a coupled photosynthesis and stomatal conductance scheme developed by Farquhar and Sharkey (1982) and Leuning (1995), respectively, and which estimates leaf-level carbon and water fluxes. Leaf-level carbon demand of C3 plants is determined by the minimum of light-limited rate ($J_e$) and Rubisco-limited rate ($J_c$), and $V_{m0}$ controls the latter as given by Eq.(1) after being scaled to a given temperature.

$$J_c = \frac{V_m(T_v)(C_{inter} - \Gamma)}{C_{inter} + K_1(1 + K_2)} \tag{1}$$

where, $V_m(T_v)$ is the maximum capacity of Rubisco to perform carboxylase function at a given temperature $T_v$ scaled from $V_{m0}$ using an exponential function (Medvigy et al., 2009) given below (Eq. 2), $C_{inter}$ is the intercellular $CO_2$ concentration, $\Gamma$ is the compensation point for gross photosynthesis, $K_1$ is the Michaelis-Menten coefficient for $CO_2$, and $K_2$ is proportional to the Michaelis-Menten coefficient for $O_2$.

$$V_m(T_v) = V_{m0} \frac{\exp\left(3000\left(\frac{1}{288.15} - \frac{1}{T_v}\right)\right)}{\left(1 - \exp\left(0.4(T_{v,lo} - T_v)\right)\right)(1 + \exp(0.4(T_v - 318.15)))} \tag{2}$$

where, $T_v$ is any given temperature for which the scaling is being done and $T_v, lo$ is the lower cut off temperature.

Stomatal conductance which is modelled using Leuning (1995), a variant of Ball Berry model (Eq. 3), is influenced by *stomatal slope* and *cuticular conductance*.

$$g_{sw} = \frac{MA_o}{(C_s - \Gamma)\left(1 + \frac{D_s}{D_0}\right)} + b \tag{3}$$

where, $g_{sw}$ is stomatal conductance for water, $A_o$ is photosynthetic rate, $M$ is *stomatal slope*, $b$ is *cuticular conductance*, $D_0$ is empirical constant, $D_s$ is water vapour deficit, and $C_s$ is $CO_2$ concentration within leaf boundary, and $\Gamma$ is as described above. Stomatal control is also affected by soil moisture supply term, which is a function of soil moisture, fine root biomass, and *water conductance*. When the available water supply is less than the demand predicted by photosynthesis-conductance model, then photosynthesis, transpiration, and stomatal conductance are all linearly weighted down to match the supply (Dietze et al., 2014).

Water and $CO_2$ concentrations within the leaf boundary layer are influenced by *leaf width* along with other factors like wind speed, leaf area index, and molecular diffusivity of heat. *Specific leaf area* (SLA) has units of leaf area per unit leaf carbon and is used to scale up leaf-level to canopy-level fluxes. Relationships between growth respiration and net photosynthesis are controlled by the *growth respiration factor*. In EDv2.2, while leaf biomass is determined by PFT specific allometric equation (as shown in Table 2 for sagebrush) based on diameter, fine root biomass is defined by a *ratio of leaves to fine roots*. *Leaf turnover* and *fine root turnover* rates together influence overall litter turnover rate, even though in deciduous trees dropping of leaves also affects this rate. Turnover rate of stored leaf pool and storage respiration depends on *storage turnover rate*, size of stored leaf pool, and storage biomass.

## 2.2 Study area

While the PFT was developed broadly for sagebrush, we developed the EDv2.2 model runs focused on the Reynolds Creek Experimental Watershed (RCEW), located in the Northern Great Basin region of Western United States (Fig. 1). The RCEW is operated by the USDA Agricultural Research Service and is also a Critical Zone Observatory (CZO) (referred to as RC-CZO). We used two 200 m x 200 m polygons centered at two EC sites within RC-CZO to closely represent the footprint area of these sites. The AmeriFlux US-Rls EC station, located at 43.1439 N and 116.7356 W and at an elevation of 1583 m, is within the Lower Sheep Creek drainage in RCEW. The footprint of this site is dominated by low sagebrush (*Artemisia arbuscula*) and Sandberg bluegrass (*Poa secunda*) (Stephenson, 1970; Seyfried et al., 2000) and is characterized as having light cattle grazing (AmeriFlux, 2018). The second AmeriFlux tower, US-Rws, is located at 43.1675 N and 116.7132 W in the Nancy Gulch drainage, about 2 km northeast of the US-Rls site. This area is dominated by Wyoming big sagebrush (*A. tridentata* ssp. *wyomingensis*) and bluebunch wheatgrass (*Pseudoroegneria spicata*) (Stephenson, 1970). Hereafter, these two sites are designated as LS (for low sagebrush) and WBS (for Wyoming big sagebrush), respectively.

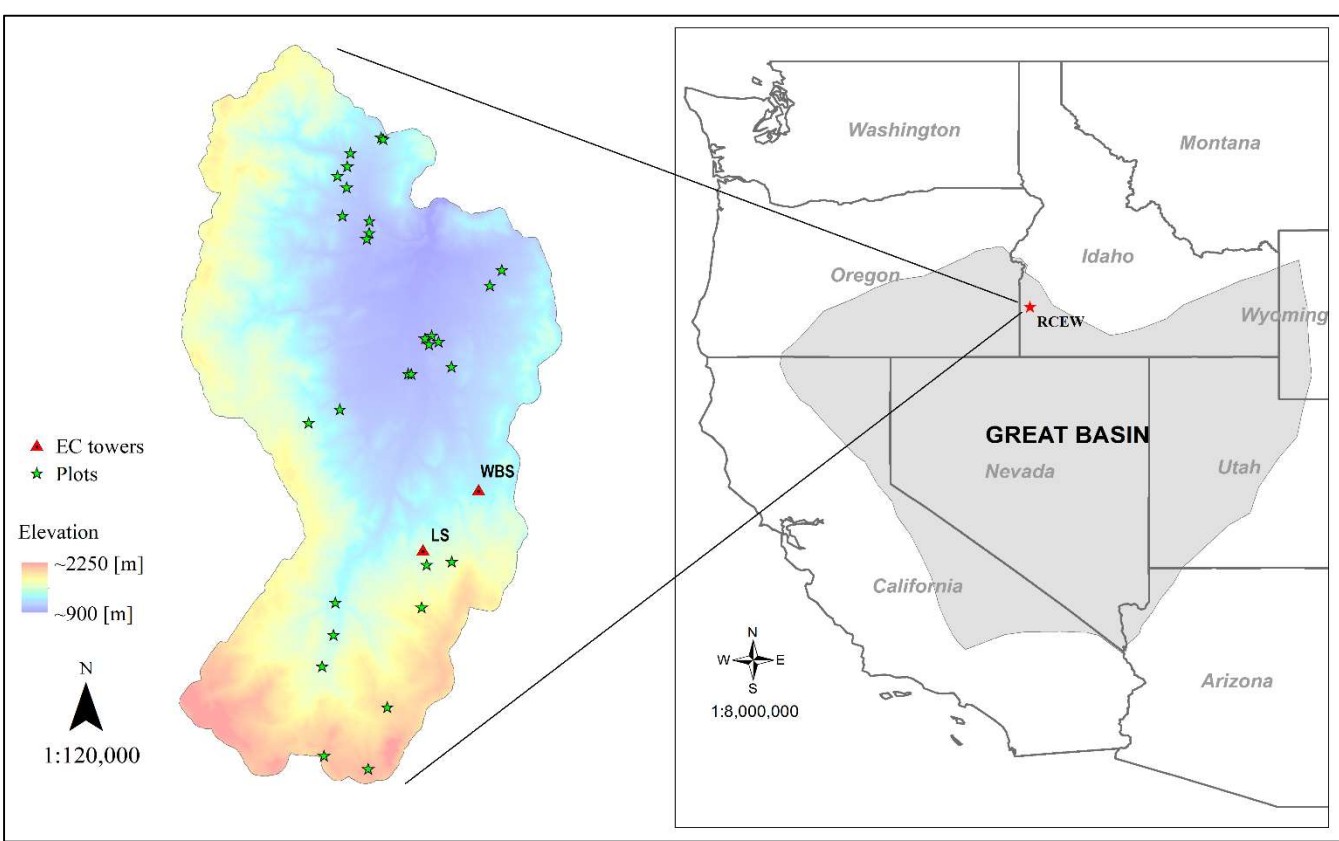

**Figure 1**. Location of EC flux sites in the RCEW study area: designated as LS (for low sagebrush) and WBS (for Wyoming big sagebrush). Field plots depicted were used to develop the allometric equations. The inset map shows the location of the RCEW study area within the Northern Great Basin (LCC, 2018).

## 2.3 Inventory and EC tower data

A field inventory dataset of sagebrush shrubs from RCEW recorded in 2014 (Glenn et al., 2017) was used to fit the allometric equations (for temperate PFTs) in EDv2.2, and to initialize the ecosystem structure for the model simulations. Variables used to fit allometric equations for the sagebrush included volume, crown diameter, height, and stem diameter. EDv2.2 was originally developed for tropical forests, and thus typically specifies allometric relationships in terms of diameter at breast height (DBH). However, this length-scale variable has limited application to shrubs of the sagebrush-steppe ecosystem, which rarely exceed 1.5 m in height. Thus, we developed a substitute length-scale variable for DBH that effectively corresponds to shrub volume. To accomplish this, shrub volume was first calculated using crown area (characterized as an ellipse, and approximated with semi-major and semi-minor axis lengths) and height, and the cube root of this volume was then used as the characteristic length-scale variable required to parameterize allometric relationships in EDv2.2. To test this relationship, we compared height predicted from cube root volume with observed sagebrush height using a different set of data from the eastern side of the Sierra Nevada Mountains, CA in the Great Basin (Qi et al., 2018). We found a good fit for the data ($r^2 = 0.71$) with a small negative bias of -1.88 cm, and a random residual distribution (Fig. S1). Using our field inventory from RCEW, we identified the coefficients in allometric equations (Table 2) for shrub height, leaf biomass, structural biomass, canopy area, and wood area index as a function of this cube-root of volume measure (used as DBH in the equation).

GPP data from 2015 to 2017 water years were derived from the LS and WBS EC stations (Fellows et al., 2017) using the REddyProc software in R (Reichstein et al., 2005) to fill and partition net ecosystem exchange (NEE) into ecosystem respiration and GPP.

**Table 2**. Coefficients for sagebrush (shrub) PFT to allometric equations in EDv2.2 (temperate PFTs).

| Relationship | Equation | Coefficients |
|---|---|---|
| DBH (cm) to Height (m) | $Ht = a(1 - e^{b \times DBH})$ | a = 4.7562, b = -0.002594 |
| DBH (cm) to Woody Biomass (kg) | $WB = \frac{a}{C2B} \times DBH^b$ | a = 5.709 x 10$^{-8}$, b=4.149 |
| DBH (cm) to Leaf Biomass (kg) | $LB = \frac{a}{C2B} \times DBH^b$ | a=2.582 x 10$^{-6}$, b=2.746 |
| DBH (cm) to Canopy Area (m$^2$) | $CA = a \times DBH^b$ | a=6.35 x 10$^{-5}$, b=2.18 |
| DBH (cm) to Volume (m$^3$) | $V = a \times Ht \times DBH^b$ | a=2.035 x 10$^{-5}$, b=2.314 |
| Volume (m$^3$) to Root Depth (m), | $D = a \times V^b$ | a = -3.0, b = 0.15 |
| DBH (cm) to Wood Area Index, | $WAI = nplant \times a \times DBH^b$ | a= 0.0096, b =2.0947 |

DBH = diameter at breast height; Ht = height; WB=woody biomass; C2B=carbon to biomass ratio; LB=leaf biomass; CA=canopy area; V=volume; D=root depth; WAI=wood area index.

## 2.4 Meteorological forcing data

Outputs from a long-term, high resolution climate reanalysis obtained from the Weather Research and Forecast (WRF) model (Skamarock et al., 2008) were used to provide meteorological forcing data for the EDv2.2 model (Table 3). The WRF outputs correspond to atmospheric temperature and specific humidity at 2 m height, wind speed at 10 m height, downward shortwave radiation and long-wave radiation at ground surface, surface pressure and accumulated precipitation (Flores, et al., 2016). The spatial and temporal resolutions of the data are 1 km and 1 hour, respectively. The EDv2.2 model then partitions shortwave radiation into direct and diffuse, visible and near-infrared components as summarized by Weiss and Norman (1985). We obtained these forcing data from 2001 to 2017 for two WRF pixels that spatially bound the LS and WBS sites (Fig. 1).

**Table 3**. Meteorological forcing data from WRF model used for simulation.

| Variable | WRF name | Unit |
|---|---|---|
| Temperature at 2 m | T2 | K |
| Surface pressure | PSFC | Pa |
| Accumulated precipitation | RAINNC | mm |
| Terrain height | HGT | m |
| U wind (zonal) component at 10 m | U10 | m/s |
| V wind (meridional) component at 10 m | V10 | m/s |
| Specific humidity at 2 m | Q2 | kg/kg |
| Downward longwave flux at ground surface | GLW | w/m$^2$ |
| Downward shortwave flux at ground surface | SWDOWN | w/m$^2$ |

## 2.5 Initial parameterization and sensitivity analysis

We identified initial sagebrush shrub PFT parameters based on field allometric equations, previous research studies on the sagebrush ecosystem (Ahrends et al., 2009; Cleary et al., 2010; Comstock and Ehleringer, 1992; Gill and Jackson, 2000; Li et al., 2009; Olsoy et al., 2016; Qi et al., 2014; Sturges, 1977; Tabler, 1964), and information from other general PFT parameters in EDv2.2 (Table S1 in the Supplement). The initial ecosystem states for the model run for the LS and WBS sites were designated to be a single sagebrush with 1 plant/m$^2$ representing average spacing from the 2014 field inventory data. For the LS site, we used 0.57 m of cube root volume (diameter) and 0.56 m for height and for WBS we used 0.62 m of cube root volume and 0.63 m for height. The soil column was configured to be 2.3 m deep with 9 vertical layers and a free-drainage lower boundary. Corresponding to a gravelly loam soil in the study site (USDA, 2018a), we used a soil texture with 55% sand, 25% silt, and 20% clay, for both sites. Initial soil moisture was set to near saturation with no temperature offset, and the initial atmospheric carbon dioxide level matching the year 2001 (370 ppm), when we initialized the simulation. We ran the EDv2.2 model with these initial settings and initial shrub PFT parameters for the sensitivity analysis at the LS site for a fifteen-year simulation period. We selected this simulation period based on our pre-sensitivity trial runs, previous studies (Medvigy and Moorcroft, 2012; Antonarakis, et al., 2014) where authors had initialized model using inventory data, and taking into account

that there have been no major disturbances in recent history in these sites. We used only one of our sites (LS site) for the sensitivity analysis because we assumed both the sites are quite similar in terms of meteorological forcing (given their proximity) and ecosystem conditions, and particularly as we used a range of maximum and minimum values of parameters in the analysis.

Since our study focused on preliminary parameterization of the sagebrush PFT, we limited the sensitivity analysis to explore linear dependence of selected parameters over target variable, assuming minimum non-linear dependence among these parameters. We used a sensitivity index (*SI*) suggested by Hoffman and Gardner (1983) (Eq. 4) to perform a one at a time sensitivity analysis and rank the parameters. Because this index is highly affected by the extreme values of parameters being studied, it is recommended that the parameter range cover the entire range of possible values. *SI* has been used in different

areas of study including ecology (Waring et al., 2016) and hydrology (Wambura et al., 2015), mostly to assess the effect of parameters on target variables, and sometimes to reduce the number of variables for further analysis.

$$SI = \frac{GPPmax - GPPmin}{GPPmax}, \tag{4}$$

where, $SI$ is sensitivity index, $GPPmax$ is the value of GPP corresponding to the simulation with the maximum value of a parameter, and $GPmin$ is the value of GPP corresponding to the simulation with the minimum value of a parameter. We identified minimum and maximum possible values for each of the selected parameters based on previous sensitivity and optimization studies, the range of parameters for other PFTs in EDv2.2, and our preliminary sensitivity analyses (Table 4). EDv2.2 was then run for a fifteen-year period with both minimum and maximum values of each parameter while keeping all

other parameters constant. The average daily GPP outputs throughout the simulation years for maximum and minimum values of parameters were used to derive $GPPmax$ and $GPmin$ respectively. We limited the optimization to the five most sensitive parameters to keep time and computing performance manageable.

### 2.6 Optimization and validation

In the third step, optimization of the five selected parameters was performed for both the LS and WBS sites using an exhaustive

search (brute force) method within the specified range of values. This process was performed to identify the best values for the five selected parameters for each EC station in predicting GPP. A Bayesian method is often preferred in parameter optimization as it can assimilate multiple input data with a single model run and provide separate uncertainties for parameters, processes, and data. However, for a model like EDv2.2, it is nearly computationally prohibitive as we would need $10^4$ to $10^7$ model runs to perform associated Markov Chain Monte Carlo processes (Dietze et al., 2018; Fer et al., 2018). Likewise, there

are model emulators (surrogate models) where statistical models are created to mimic full models by fitting parameters and response variables using distributions such as Gaussian. Experiments done with these model emulators are later transferred into the full model thus making this method computationally efficient. One of the drawbacks of this method is that it frequently

fails to converge with non-linear parameters (Fer et al., 2018; Keating et al., 2010). An alternative to these approaches is the brute-force method where all possible combinations of parameters from a uniform distribution within a pre-defined range are examined to get the best result. Advantages of the brute-force method are a higher possibility of identifying global optimums or fine tuning of posterior parameter ranges and assessing non-linearity among parameters. The major disadvantage of this method is the computational cost but this can be reduced significantly by limiting the range of parameter domain (Fer et al., 2018; Schmidtlein et al, 2010).

For each site, we ran 720 simulations with a unique combination of parameter values for fifteen years (2001-2016), at which point it was assumed to reach an equilibrium with climate. EDv2.2 simulations were configured to allow for growth of the C3 grass, northern pines, and late conifers together with the shrub PFT. This was done because although the vegetation assemblages in the flux site footprints are primarily composed of sagebrush and grasses, conifers are present in some parts of the experimental watershed (Seyfried et al., 2000). For each simulation, we calculated a skill score, Nash-Sutcliffe efficiency (NSE) (Nash and Sutcliffe, 1970), to compare the simulated GPP from 2015 and 2016 with those derived from LS and WBS EC stations for respective years. Although, NSE is closely related to root mean square error (RMSE) (or mean square error, MSE), the skill score from it can be interpreted as comparative ability of the model over a baseline model, which is the mean of site observations in this case. While the RMSE value depends on the unit of predicted variables, which can vary from 0 to infinity, the NSE is dimensionless and varies from negative infinity to 1 (Krause et al, 2005; Gupta et al, 2009). NSE is calculated using Eq. (5):

$$NSE = 1 - \frac{\sum_{i=1}^{n} (O_i - P_i)^2}{\sum_{i=1}^{n} (O_i - \underline{O})^2}, \tag{5}$$

where, $O_i$ is observation, $P_i$ is predicted value, $\underline{O}$ is mean of observation, and $n$ is number of observations. For both EC stations, we selected the ten best simulations based on NSE scores, computed ensemble means of all five parameter values, and estimated mean GPP. Outputs from process-based models like EDv2.2 are often ill-posed, meaning that there may not be a unique solution of parameter combinations but rather several combinations of parameters produce the same solution. One way to solve the ill-posed problem is by selecting more than one of the best combinations, from which we can either explore average outputs or select one of the ensemble members that would better match any prior information such as any correlation among parameters, available data, vegetation characteristics or ecosystem conditions (Combal et al., 2002; Quan et al., 2015). The simulated GPP from these runs were then compared against respective EC site data from 2017, which was withheld from the optimization as a means of providing an independent validation.

## 3 Results

### 3.1 Initial parameterization and sensitivity analysis

For the model run based on the initial values of parameters (Table S1 of Supplement), the fifteen-year simulations produced an annual cycle in GPP that decreases in amplitude during the initial 1-3 years, and remains at a level of approximately 0.07 kgC/m$^2$/yr in the remaining years (Fig. 2a). Observed GPP in 2016 were 0.51 kgC/m$^2$/yr and 0.38 kgC/m$^2$/yr for the LS and WBS sites, respectively. This result was significantly lower than the observed GPP from either of the EC sites, and thus we followed up with sensitivity and optimization analysis to constrain some of the influential parameters.

Based on the SI ranking, SLA, stomatal slope, $V_{m0}$, fine root turnover rate, and Q-ratio were identified as the top five sensitive parameters compared to the other parameters explored (Fig. 2; Table 4). Related studies (Dietze et al., 2014; Medvigy et al., 2009; Pereira et al., 2017; Zaehle et al., 2005) have also identified similar model parameters being important in estimating GPP. In our study, higher parameter values of SLA, stomatal slope, and $V_{m0}$, resulted in higher GPP estimates (Fig. 2b, c, and d), whereas for Q-ratio and fine root turnover rate, higher parameter values produced lower GPP (Fig. 2e and f). The impact of shifts in SLA, $V_{m0}$, and stomatal slope values are observed from the very beginning of the simulations, while changes in fine root turnover rate and Q-ratio parameters start to show differences from roughly 3-4 years after the initial model run. Although not ranked in the top five, cuticular conductance, leaf turnover rate, and growth respiration factor also had considerable influences over GPP (Table 4).

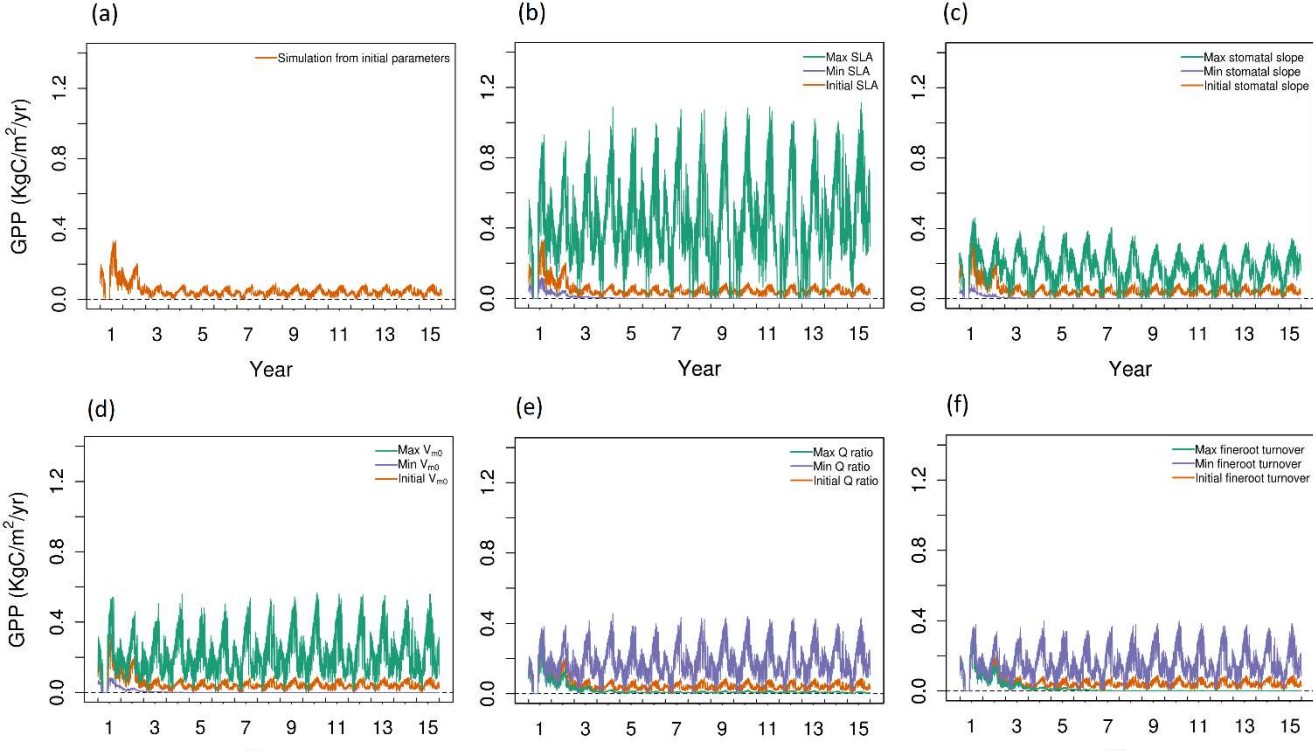

**Figure 2**. Simulated daily GPP outputs from 1-15 years for the study location with (a) initial values of all five parameters, and (b-f) maximum (green), minimum (blue), and initial (red) parameter values for SLA, stomatal slope, $V_{m0}$, Q-ratio, and fine root turnover rate.

**Table 4**. Summary of sensitivity analysis of studied PFT parameters ranked by Sensitivity Index (SI).

| Parameters | Initial | Min | Max | SI | Reference |
|---|---|---|---|---|---|
| SLA ($m^2kg^{-1}$) | 4.5 | 2 | 15 | 0.988 | Lambrecht et al., (2007); Brabec (2014); Olsoy et al., (2016) |
| Stomatal slope | 7 | 2 | 15 | 0.983 | Dietze et al., (2014); Bonan et al., (2014) |
| $V_{m0}$ ($\mu molm^{-2}s^{-1}$) | 16.5 | 4 | 30 | 0.982 | Comstock & Ehlenger (1992); Oleson et al., (2013) |
| Q-ratio | 3.2 | 0.4 | 12 | 0.898 | Dietze et al., (2014) |
| Fine root turnover rate ($a^{-1}$) | 0.33 | 0.1 | 2 | 0.895 | Gill and Jackson, (2000) |

| | | | | | |
|---|---|---|---|---|---|
| Cuticular conductance ($\mu molm^{-2}s^{-1}$) | $10^3$ | $10^2$ | $10^7$ | 0.813 | Barnard and Bauerle (2013): Duursma et al., (2018) |
| Leaf turnover rate ($a^{-1}$) | 1 | 0.1 | 2 | 0.779 | * |
| GRF | 0.33 | 0.11 | 0.66 | 0.694 | Wang et al., (2013) |
| Water Conductance ($ms^{-1}kgCroot^{-1}$) | $1.9 \times 10^{-5}$ | $1.9 \times 10^{-6}$ | $1.9 \times 10^{-4}$ | 0.168 | * |
| Storage turnover rate ($a^{-1}$) | 0.624 | 0.33 | 0.95 | 0.004 | * |
| Leaf width (m) | 0.05 | 0.01 | 0.10 | 0.002 | * |

* Information about the range comes from range of values for other PFTs in EDv2.2, and our preliminary sensitivity analysis

### 3.2 Optimization and validation

For our exhaustive search of parameter values, we limited search domains for parameters based on previous studies and the result of our sensitivity analysis. SLA search limits were largely based on Olsoy et al. (2016), who suggested a range of 3 to 6 $m^2$/Kg for sagebrush SLA, with regional and seasonal variations. Similarly, limits for $V_{m0}$ were extended slightly beyond Comstock and Ehleringer's (1992) recommendations for Great Basin shrubs, and the upper limit for stomatal slope was extended slightly beyond that used by Oleson et al. (2013) for a shrub PFT in the Community Land Model (CLMv4.5). We set search domains for Q-ratio based on a leaf and root biomass study of sagebrush by Cleary et al. (2010), and fine root turnover ratio was based on results from a study on *Artemisia ordosica* in a semi-arid region of China (Li et al, 2009). Interval distances (or 'steps') were calculated to equally space out the range between the maximum and minimum of each parameter for a given number of intervals (Table 5). Parameters identified as exerting more control on GPP prediction were assigned higher number of steps, resulting in the following: five steps of SLA, four steps for $V_{m0}$, stomatal slope, and three steps for Q-ratio and fine root turnover rate. Among 720 possible simulations for unique parameter value combinations for each site, 92 cases from LS and 116 cases from WBS which did not provide model optimization results because of numerical instabilities (with GPP approaching zero) were excluded from subsequent analysis.

**Table 5**. Minimum value, maximum value, interval size, and number of steps for each parameter used in optimization.

| Parameter | Min | Max | Interval | Number of steps |
|---|---|---|---|---|
| SLA ($m^2kg^{-1}$) | 3.00 | 9.00 | 1.50 | 5 |
| $V_{m0}$ ($\mu molm^{-2}s^{-1}$) | 14.00 | 21.50 | 2.50 | 4 |
| Stomatal slope | 7.00 | 10.00 | 1.00 | 4 |
| Fine root turnover ($a^{-1}$) | 0.11 | 0.33 | 0.11 | 3 |
| Q-ratio | 0.40 | 3.20 | 1.40 | 3 |

We selected ten simulations with the best NSE scores for both the LS and WBS sites (Table S2 and Fig. S1 in the Supplement) and determined ensemble means of parameter values for these sites (Table 6). To perform validation of these ten best simulations from each EC station, we extended the model runs to obtain GPP estimates for the year 2017. We then

compared the biases and skill scores associated with the top performing simulation (hereafter the 'best case') and the mean from all ten simulations (hereafter the 'ensemble mean'). Among the ten best simulations selected for each EC site, four of them were common to both sites (Table S2 in the Supplement). We observed that the variation in parameter values was more pronounced for the LS site, especially with regard to $V_{m0}$ and stomatal slope. Likewise, we identified more variation in GPP estimates among ten best simulations for LS site than for WBS site, especially during the peak and trough periods in the plots (Fig. S2 in the Supplement). The best case for WBS site showed traces of C3 grass growth through some intermediate simulation years even though we initialized the model with only the shrub PFT (Fig. S3 in the Supplement). Optimized parameter values were only slightly different between the best case and ensemble means for both sites, possibly suggesting little interaction effects among the parameters (Table 6). In the best case, parameter values for $V_{m0}$, SLA and stomatal slope were the same for both the sites, whereas Q-ratio and fine root turnover rate were different. We also observed that fine root turnover rate and Q-ratio had higher variability among the ten best simulations compared to the rest of the parameters for both sites. Our comparison of variation between the two sites among ensemble members showed the WBS site had overall lower variation than the LS site (Table 6), and $V_{m0}$ had noticeably lower variation for the WBS site.

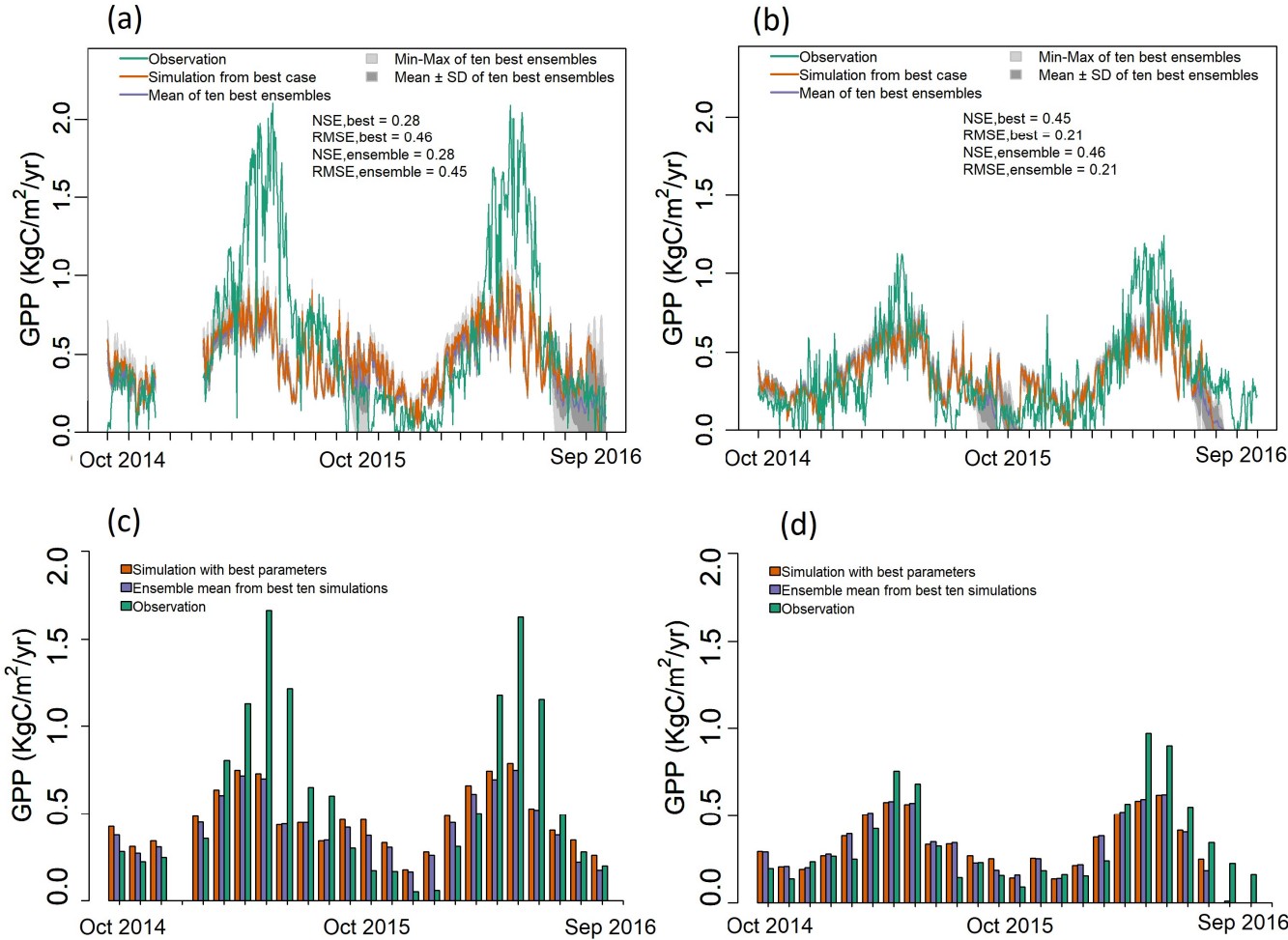

**Figure 3**. Observed and simulated GPP for the optimization period (water years 2015 and 2016) for both EC towers. a-b. Simulated daily GPP (kgC/m²/yr) from best case, one standard deviation of ensembles, and range of ensemble simulations compared with observation for a) LS and b) WBS sites. c-d. Simulated mean monthly GPP (kgC/m²/yr) from best case and ensemble mean compared against observation data for c) LS and d) WBS sites. Note that observation data from December 11, 2014 to February 17, 2015 was missing for LS site.

Figure 3 presents simulated GPP from the best case, variation among ten best ensemble members, and ensemble mean for the final two model years (October 2014 to September 2016) along with the observed GPP from the same period from each EC station. Optimization results for the LS site in Fig. 3a show that simulated GPP matches well with observed data for most days, except during the spring season, during which strong peaks in observed GPP were not captured by the simulation results. Average monthly comparisons (Fig. 3c) show that simulated GPPs are close to the observations for most of the months except for April, May and June, during when the model is clearly underestimating GPP. We observed a small variation (Fig. 3a) among ten ensemble simulations with an average standard deviation of 0.057 kgC/m²/yr where most of the variations were observed during fall and early spring. Variations during fall months are evident in the monthly average GPP, in which there

was often considerable difference between ensemble mean and best case estimations for September and October (Fig. 3c and 3d). Despite GPP estimation from ensemble mean (-0.17 kgC/m$^2$/yr) having higher negative bias compared to the best case (-0.14 kgC/m$^2$/yr), its skill score (NSE) was marginally higher (Table 7). In comparison to the LS site, the WBS site had lower spring peaks in GPP, which were also limited to fewer months (Fig. 3b and 3d) and were far more comparable to the simulation

results. The average standard deviation among ensemble simulations (0.037 kgC/m$^2$/yr) was lower for WBS than for LS, resulting in little difference between best case and ensemble mean estimations for that site. Both the WBS and LS ensemble mean simulations produced only a marginally higher NSE than the best case results. However, the spring mismatch in the LS site resulted in higher Bias and lower NSE when compared to the WBS site (Table 7). Yet, despite negative biases during spring, positive NSE scores for both sites suggest that the parameters were generally functioning to allow the model to track

observed daily GPP over time.

**Table 6**. Optimized parameter values from best cases and top ten ensembles (mean and standard deviation (SD)) for LS and WBS EC stations.

| Parameters | LS EC station | | WBS EC station | |
|---|---|---|---|---|
| | Best case | Ensemble mean and SD | Best case | Ensemble mean and SD |
| $V_{m0}$ (µmolm$^{-2}$s$^{-1}$) | 19.00 | 19.00 ± 1.66 | 19.00 | 18.25 ± 1.21 |
| SLA (m$^2$kg$^{-1}$) | 7.50 | 8.10 ± 0.77 | 7.50 | 8.10 ± 0.77 |
| Stomatal slope | 9.00 | 8.70 ± 0.94 | 9.00 | 9.10 ± 0.32 |
| Fine root turnover (a$^{-1}$) | 0.22 | 0.19 ± 0.09 | 0.33 | 0.23 ± 0.08 |
| Q-ratio | 3.20 | 2.08 ± 1.29 | 1.80 | 1.80 ± 1.14 |

For validation of the parameter estimates, we ran the EDv2.2 model for all ten best simulations with corresponding parameter values from 2016 to 2017 for both the LS and WBS sites and compared the simulated GPPs from the 2017 water year with observed GPP from the respective locations in the same year. Results from the model validation showed greater negative biases and lower NSEs for both sites compared to the optimization results (Table 7). Moreover, there were substantial differences in mean GPP observations from EC sites for both LS and WBS sites, between optimization (LS = 0.61 kgC/m$^2$/yr,

WBS = 0.35 kgC/m$^2$/yr) and validation (LS = 0.55 kgC/m$^2$/yr, WBS = 0.46 kgC/m$^2$/yr) years. Validation results were slightly better for the WBS than the LS site, however, the difference in validation performance among the two sites was not as distinct as with the optimization results. Overall, positive NSE values for both cases (best-case and ensemble mean) for both sites suggest the simulated estimates provided better GPP predictions than the observed means. Poor validation results could be attributed to inter-annual variability in observed GPPs and to the inability of the model to adequately capture peak spring

growth.

**Table 7**. Bias, NSE, and RMSE for optimization and validation of GPP for the best case and the top ten ensemble mean for both EC stations.

| Simulations | Optimization | | | Validation | | |
|---|---|---|---|---|---|---|
| | Bias (kgC/m$^2$/yr) | NSE | RMSE (kgC/m$^2$/yr) | Bias (kgC/m$^2$/yr) | NSE | RMSE (kgC/m$^2$/yr) |
| *LS* | | | | | | |
| Best case | -0.137 | 0.277 | 0.456 | -0.257 | 0.069 | 0.554 |
| Ensemble mean | -0.172 | 0.281 | 0.455 | -0.278 | 0.053 | 0.559 |
| *WBS* | | | | | | |
| Best case | -0.028 | 0.452 | 0.213 | -0.252 | 0.079 | 0.411 |
| Ensemble mean | -0.030 | 0.456 | 0.212 | -0.265 | 0.036 | 0.420 |

Validation results shown in Fig. 4 also indicate that the simulated daily GPPs for both sites matched observed values relatively well from late fall until early spring months (October to April), but performed poorly in the late spring and summer months (May to September) when compared with observation data from 2017. Daily patterns of simulated GPP were almost identical for both sites, with GPP falling sharply through late summer months and remaining close to zero. Observed GPP data values at both sites showed similar patterns of decline in 2017 during late summer months (July and August), though not as sharply as the simulated results (Fig. 4). The observed increase in GPP at the beginning of fall (September) was not well captured by the simulated outputs for either site. Monthly averages also clearly show differences between simulated and observed GPP for May through September (Fig. 4 c & d). Variation among ensemble simulations was higher for the LS site compared to the WBS site, with standard deviation of 0.056 kgC/m$^2$/yr and 0.02 kgC/m$^2$/yr, respectively. When validation and optimization results are compared, the variation between the ensemble simulations of the LS site were relatively similar, whereas ensemble variation was generally lower in the validation output for the WBS site. Ensemble means for both sites for validation exhibited almost identical patterns as the best case simulations, though at slightly lower levels for most of the months.

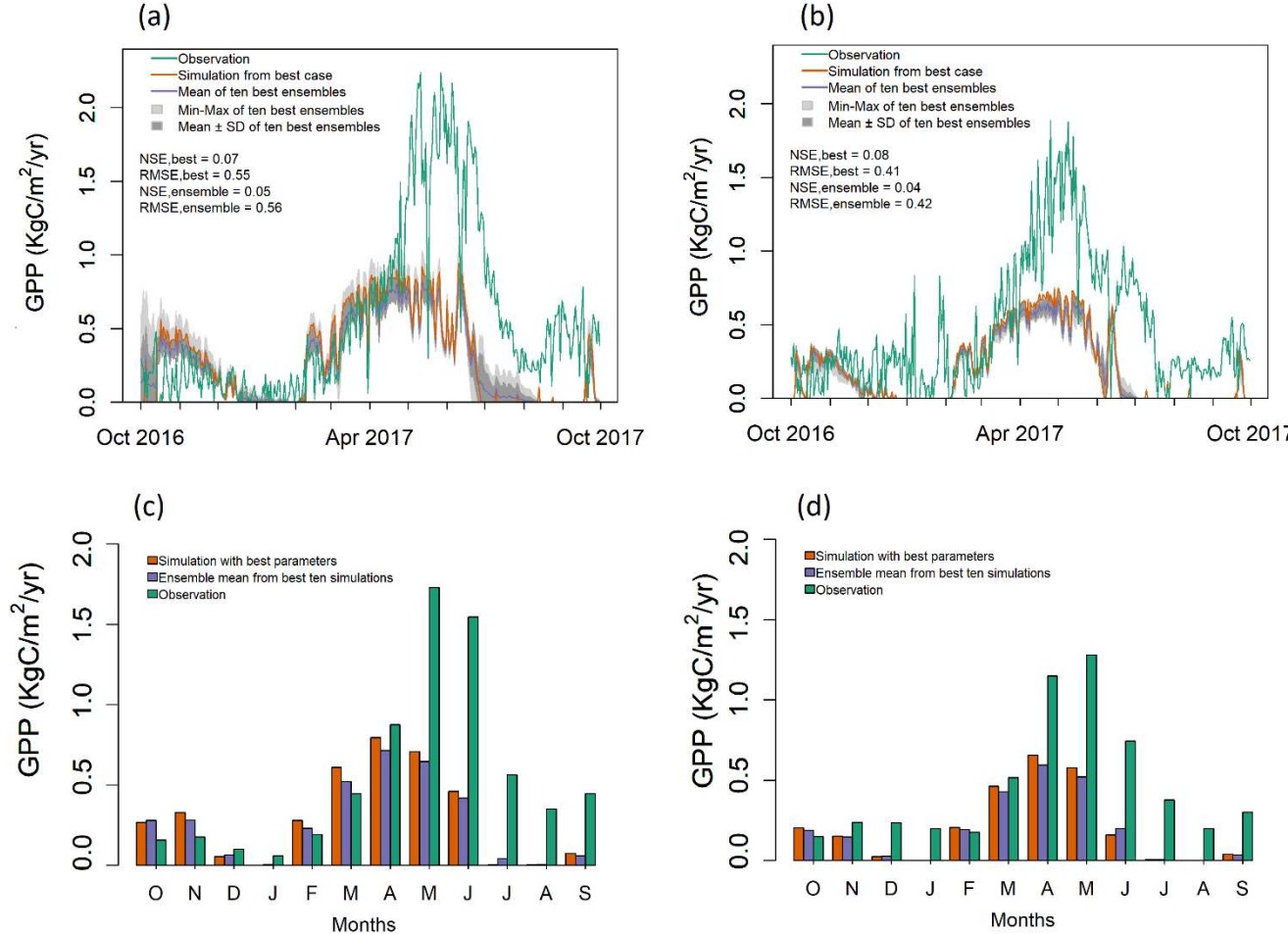

**Figure 4.** Observed and simulated GPP for validation period (water year 2017) for both EC towers. a -b. Simulated daily GPP (kgC/m$^2$/yr) from best case, one standard deviation of ensembles, and range of ensemble simulations compared with observation for a) LS and b) WBS sites. c-d. Simulated mean monthly GPP (kgC/m$^2$/yr) from best case and ensemble mean compared against observation data for c) LS and d) WBS sites.

## 4 Discussion

Using our newly developed sagebrush shrub PFT, we were able to effectively simulate sagebrush ecosystem productivity in EDv2.2 as represented by the two study sites. Simulated results, after about four modelled years, clearly maintained annual shrub GPP over time, although at a lower level than the observed data from these sites. To improve GPP estimates and reduce uncertainty, we assessed sensitivity of eleven different parameters closely associated with biomass growth. Results from this preliminary analysis were similar to previous studies (Dietze et al., 2014; LeBauer et al., 2013; Medvigy et al., 2009), wherein parameters $V_{m0}$, SLA, fine root turnover rate, and stomatal slope were found to be the most influential in determining carbon flux or primary productivity. We observed variation in these parameter values for ten best simulations selected based on NSE

values (Table S2 in the Supplement) for both sites where the LS site had higher variation in $V_{m0}$ and stomatal slope than the WBS site. The effects of some parameters (stomatal slope, fine root turnover rate, and Q-ratio) on GPP prediction differed when they were altered individually versus simultaneously with other parameters. For instance, sensitivity analysis suggested GPP increases when fine root turnover ratio and Q-ratio are lowered individually, yet the best results for each site did not

improve (i.e., still under-predicted GPP) with the lowest values of these parameters. Indeed, in addition to first order effects of the studied parameters, the top ten best parameter combinations exhibited variation in parameter values for both EC sites, suggesting interacting effects and potential nonlinear dependence among parameters. The best case parameters identified from the optimization suggested some difference in fine root turnover rate and Q-ratio between the LS and WBS sites. We found a similar pattern of marginally higher fine root turnover rate and lower Q-ratio at WBS with the ensemble mean, despite

substantial variation among the ten ensemble members. We can potentially relate these differences in parameters between the LS and WBS sites with differences in the root systems and vegetation height of low sagebrush and Wyoming big sagebrush species which are the dominant vegetation types of the respective sites. Low sagebrush is smaller plant with primarily a shallow fibrous root system whereas Wyoming big sagebrush is taller with a dual tap root and shallow root system (Steinberg, 2002; USDA, 2018b).

Negative bias in estimated GPP for the best simulations resulted from an inability of the model to correctly produce daily GPPs for late spring and summer months. Although a higher annual GPP could be obtained to compensate for negative bias by changing parameters values, the highest GPP was not necessarily the one with the best NSE, since NSE was calculated based on daily GPP values. Limiting optimization to five of the eleven parameters initially identified may have also contributed to the error and bias observed in our modelled estimates. Our generalization of shrubland ecosystem processes (as trees in the

EDv2.2 model structure) may also be one of the limiting factors. Open and scattered shrubland ecosystems as in our study area (Mitchell et al., 2011) do not follow the same pattern of recruitment and competition as we would expect for a closed canopy ecosystem (Schwantes et al., 2016; Wolf et al., 2008). In addition, future studies should give special consideration to phenology, seed dispersal, and mortality that are unique to these shrubland ecosystems. Even though the PFT parameters and allometric coefficients that we developed for shrub PFT influence the above-mentioned ecosystem processes, we suggest

modification of some of these model structures in future studies to test their influence and potentially improve GPP estimates.

    GPP simulations for the WBS site had better optimization scores than for the LS site, and also a slight edge over the latter for validation results. This could be due to differences in soils and hydraulic conditions between the sites as we used similar setups for our simulation. Moreover, variation between morphological characteristics of the vegetation at the LS and WBS EC towers (characterized by low sagebrush and Wyoming big sagebrush, respectively), including growing seasons and flowering

seasons, may also have resulted in the observed differences in GPP (Howard, 1999; USDA, 2018b). Since Wyoming big sagebrush is the dominant species in the Reynold Creek Watershed area (Seyfried et al., 2000), the allometric equations fitted for sagebrush (representing most areas of RCEW), could favor the more realistic growth pattern of this species in the model (e.g., Fig. 3 and 4).

Additionally, differences in the phenology of the associated grass species between the two sites could result in differences in seasonal and annual productivity (Cleary et al., 2015). For instance, the perennial grass at the LS site is Sandberg bluegrass, which is photosynthetically active in early spring and senesces by early summer (USDA, 2016), and thus may have contributed to the observed higher spring GPP peak at the LS site. Although, we observed small amounts of simulated GPP growth for C3 grasses for certain intermediate years, these levels were not sustained. However, current parameters for C3 grasses were unlikely to adequately produce co-existence of grasses in the area, and we could not validate results in terms of the actual species composition and ecosystem dynamics of the EC sites, as we did not have GPP observations for unique PFTs. We also observed high inter-annual variation in observed GPP for both sites, leading to poor results in validation of simulation outputs. In summary, site-specific variability, model complexity, and optimizing for only five parameters likely contributed to, or were responsible for, the differences between modelled and observed GPP estimates.

While the emphasis of this study was to develop and optimize the shrub PFT parameters, we expect that simultaneous optimization of both grass and shrub PFTs would result in improved representation of the vegetation composition in the study area. Such an effort would also increase the number of parameters required, potentially complicating the process of optimization and validation unique to each PFT. Moreover, several studies suggest that the parameters $V_{m0}$ and SLA vary considerably across seasons (Groenendijk et al., 2011; Kwon et al., 2016; Olsoy et al., 2016; Zhang et al., 2014). The mismatch in daily GPP patterns between simulated and flux tower data for specific seasons could be partly attributed to the lack of the model's ability to address these seasonal deviations correctly. Like most other terrestrial biosphere models, EDv2.2 does not incorporate seasonal variation in $V_{m0}$, SLA, or other model parameters (Medvigy et al., 2009).

The optimization scheme implemented in our study has some limitations. For example, we assumed the distribution of all parameters of interest to be uniform while this may not be true. Similarly, to keep computational time practical, we excluded some of the sensitive parameters such as cuticular conductance, leaf turnover rate, and GRF from the optimization analysis. We may achieve better results in parameter optimization and GPP estimates by making advances in our methods in future studies. For example, we can utilize additional sensitivity (including variance decomposition, first order and second order analysis) (Zhang et al., 2017) and optimization (including cost function, gradient descent, and uncertainty analysis) (Richardson et al., 2010) methods to fine tune the sagebrush PFT parameters. Similarly, if we include additional years of observation data, we may better capture interannual variability normally observed in ecosystem fluxes, and potentially improve validation outcomes.

## 5 Conclusions

This study demonstrates that despite the complexity of the sagebrush-steppe ecosystem, estimating GPP using the newly developed sagebrush PFT is comparable, although with seasonal-bias, to observations obtained from EC station sites. Since our primary focus here was to develop initial parameters (including allometric relationships) for the shrub (sagebrush) PFT in EDv2.2, we focused our efforts on utilizing simple sensitivity and optimization tools to constrain errors associated with

simulated GPP. Our identification of coefficients for allometric equations coupled with the other parameters for the semiarid shrub PFT for EDv2.2 will permit exploration of additional research questions. For instance, EDv2.2 could be run at regional scales with optimized parameters to model the spatiotemporal dynamics of the sagebrush community composition and ecosystem flux, under different climate and ecological restoration scenarios. PFT parameters identified and constrained from this study can be used as preliminary prior information in future studies related to sagebrush. We can either use the best case parameter sets from one of the study sites, depending upon the dominant sagebrush type, or we can use any one of the ten ensemble parameters if we have reliable information on the studied parameters. With additional time and computing resources (to facilitate large numbers of simulations), we can further refine sagebrush parameters to explore variance decomposition and non-linear dependencies using different sensitivity and optimization methods. Optimization of associated or co-occurring PFTs (C3 grass and conifers) in the region spanning out to include additional study sites, would also help to better understand and constrain uncertainties in estimating the complex dynamics of the sagebrush-steppe ecosystem.

*Code & data availability.* Original EDv2.2 is available at Github (https://github.com/EDmodel/ED22), which is maintained and continuously updated by the owners of the repository. Modified source codes for EDv2.2 with shrub PFT parameters used in this paper and input data are available at https://doi.org/10.5281/zenodo.2631988 (Last access: 08 April, 2019).

*Author contribution.* KP led the model simulation and manuscript preparation with significant contributions from all co-authors. KP, HD, NFG, ANF, KCM, and DJS conceived the idea and contributed in research design. KP, KCM, and HD led work on fitting shrub allometric equations and sagebrush parameters, with feedback from all other authors. GNF and AWF processed EC tower data to use in the analysis.

*Competing interests.* The authors declare that they have no conflict of interest.

*Acknowledgements.* This work was supported by the Joint Fire Science Program grant (Project ID: 15-1-03-23), NASA Terrestrial Ecology (NNX14AD81G), and NSF Reynolds Creek CZO project (58-5832-4-004). We thank Nayani T. Ilangakoon and Lucas P. Spaete from Boise Center Aerospace Laboratory (BCAL), Boise State University; Dr. Matt Masarik from Lab for Ecohydrology and Alternative Futuring (LEAF), Boise State University; and the staff at Research Computing, Boise State University. Part of this work was performed at R2 computer cluster, Boise State University. Authors are thankful to Dr. Paul R. Moorcroft, Center for the Environment, Harvard University; Dr. David Medvigy, Department of Biological Sciences, University of Notre Dame; Dr. Ryan G. Knox, Earth Science Division, Lawrence Berkley National Laboratory; and Dr. Michael Dietze, Earth and Environment Department, Boston University for valuable inputs in exploring shrub PFT parameters. We also thank Dr. Matthew Germino and Dr. David Barnard for expert review of the input parameters and review of the manuscript, respectively. Any use of trade, product, or firm names is for descriptive purposes only and does not imply endorsement by the U.S. Government.

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
