# Peer review of "Developing and optimizing shrub parameters representing sagebrush (*Artemisia* spp.) ecosystems in the Northern Great Basin using the Ecosystem Demography (EDv2.2) model"

_Geoscientific Model Development, 2018_

## Referee Comment (RC1) · Anonymous Referee #1 · 8 Jan 2019

The manuscript tries to provide a new parameter set for the representation of shrubs in the ED2 – DGVM. The implementation aims to improve GPP estimation in shrublands.

Yes, shrublands are under-represent in DGVMs and need more consideration, but I think the present manuscript need an extensive revision to show that shrublands work well within the ED2 model. For two sites a simple methods is used to optimise the parameter values, but the study provide no cross-validation and no further application is given.

As I have general caveats about the methods used in this study I will list them here and will not go into much detail.

1. Most importantly, the method used here to optimise parameters is not state of the art. There are a lot of methods usually applied to solve the problem of parameter optimisation as the Monte Carlo Analysis or genetic optimisation algorithms. Then it would be possible to include all important parameter for the optimisation procedure. 2. Secondly, the same as for the parameter optimisation, the parameter sensitivity measure should be performed with a more comprehensive method (e.g. using partial rank correlation coefficient (PRCC) or Fourier Amplitude Sensitivity Test (FAST)). A freely available paper ( https://www.ncbi.nlm.nih.gov/pmc/articles/PMC2570191/ ) gives a overview of the methods, which can be used to conduct parameter optimisation and sensitivity tests. 3. Another point is that the authors should use both sites to optimise the parameter set, if they want to apply the model on a broader scale. Furthermore, I didn't understand why the study provides the 10 best ensemble means, these can't be better than the best estimate. But anyhow the authors don't provide a cross-validation. Hence it is impossible to evaluate the performance of the optimised parameters as these are used for the optimisation already.

Some other important points are striking:

Metrological data are used for a different time period as the GPP data to which parameters are optimised. If you perform a parameter optimisation specifically for a site, you should use the metrological data of this site, which are normally provided by the EC tower data. But at least the same time period needs to be used. The authors state that the equilibrium is reached after 15 years, which seems to be very short. Figure 2 gives a hint that equilibrium is maybe not reached. It is not clear to me if the ED2 model used here includes the nitrogen cycle or if the fire dynamics is turned off for the optimisation procedure. It is strongly stated in the introduction that fire dynamics plays an important role in the global carbon balance, but isn't treated in the study! Authors mentioned that they have changed the allometric equations, but it is never written how,

please add that to your manuscript as it is an important information. But also how the used parameter are applied in the model would be a nice additional information. This would help the reader to understand why parameters are sensitive or maybe not. Why do you use a different parameter range for optimisation and sensitivity test, or did I get it wrong? And how did you define the parameter range? I missed some references here. The TRY database is an extensive source to determine the parameter range. You have not shown any measures in the figures. And I do not agree that it is a good match for a site-specific optimisation as stated in the manuscript. Lastly, there are a lot of statements in the abstract and in the introduction about the global importance of shrublands for the global carbon cycle, but authors don't show an application.

---

## Referee Comment (RC2) · Anonymous Referee #2 · 11 Jan 2019

Review of: Optimizing shrub parameters to estimate gross primary production of the sagebrush ecosystem using the Ecosystem Demography (EDv2.2) model By:Karun Pandit et al.

General comments This article is about the optimization of sagebrush parameters based on GPP in the EDv2.2 model and in Great Basin.

The development and optimization of specific vegetation - here a specific shrub - are currently a key research area to increase model adequacy with observations and enable the simulation of the future development of our ecosystems. However, contrary to that suggested by the actual title, in this article there is no presentation of GPP results estimation but rather only some "optimization and validation" of parameters. Moreover, the article does not present some general case but more a specific situation: a very small zone (200m2 simulated, 2 years of observation in 2 points). I suggest to change the title to make it more explicit. The model used here is EDv2.2, which seems interesting for medium scale simulation. But the methods used limit the scope (and so the interest) of this study. The two differents sites of observation are located close to each other but differ in the type of sagebrush present (a small specie and a big specie). However, only one allometry parameterization is proposed. Your choice to have a dynamic vegetation is curious considering you work on two very specific and well documented sites for one unique year (one for optimization and one for validation). Some of the methods used (as the use of sensitivity index) rely on strong hypotheses, that have been presented only in the discussion. Some deeper bibliography could have made it possible to anticipate errors. The purpose of the article to estimate GPP is thus more a local application of the optimization of parameters in order to simulate (not here) the GPP. Due to the small data set and the validation performed without any statistical test (and one of the two cases that seems not so adequate), there is no insurance that the method could be applied for other years (to predict) or in other sites. As no specific development was presented here, except an adaptation of parameters for sagebrush allometry, the relevance of this article for publication in GMD can be questioned.

Globally, considering the 14 detailed comments presented below, the editorial and figure quality of the present manuscript, I consider that in this state this article lacks of consistency and does not reach the standard quality expected for GMD.

Specific comments

1) Not only simulations or field observations can be used to quantify GPP (p.1 l.6). A third essential data set comes from satellites and remote sensing, providing continuous values (spatially and over time). There is for example the GPP

from the FLUXCOM project (Tramontada et al., 2016 https://doi.org/10.5194/bg-13-4291-2016 and Jung et al, 2017 https://doi.org/10.1038/nature20780) or from a linear relationship with the Sun-Induced Fluorescence (Su et al., 2017 http://resolver.caltech.edu/CaltechAUTHORS:20171016-145548969). Of course the problem of isolating the GPP for a PFT remains. . . as is the case for the observations used in this study. Moreover, this GPP data can be used (if you know the vegetation distribution) to do more efficiency optimization and/or validation (largest temporal and/or spatial scale).

2) As indicated in the article (p.2 l.19), it could be difficult in models to represent and parameterize specific ecosystems and they are historically not well simulated. But this is currently a major point of development in land surface models, as for tundra (mosses, shrubs,. . .) which are now more and more represented. The sentence "Semi-arid, non-forest ecosystems provide an excellent example of this limitation" (p.2 l.20) has to be more documented. More generally, a short review of the current state of what is done in different models would be necessary in this article. Nevertheless, it is probable that these models are not yet sufficient to reproduce specifically the sagebrush.

3) Globally all the references of the article have to be checked. There are wrong dates in the reference list (e.g. for Bradley and Chambers), some references are missing (e.g. Skamarock et al, 2008 and Wright et al., 2004), others are never used in the text (e.g. Brabec et al,2001 and NPS, 2018) and one seems wrong (Davidson et al., 2011 about amazon forest to illustrate tundra). You also have two undifferentiated "USDA, 2018".

4) In the introduction (as suggested in the title) you say that you are going to predict the GPP (p.3 l.4). This seems a little ambitious compared to what is actually done in the result section: an optimisation and validation. In my sense, prediction consists in running the model in the future and simulating the future evolution of GPP.

5) At the beginning of the methods (p.3 l.16 to 23), you are doing a distinction between

two types of model: "gap" or "big leaf". If the general differentiation between both is clearly understandable, some inaccuracies have to be checked and the references have to be improved / updated. (a) p.3 l.20 and l.22 you indicate that in individual-based models you can have competition, coexistence and disturbance, and that it is a limit for the big leaf model. But you have also big leaf models (DGVM) with competition, disturbance,... (b) p.3 l.23 you indicate that individual-based models have problems due to computing cost, but this is becoming less and less of a problem and currently many large-scale models (initially big leaf) have developed individual based version. Moreover, in this article the small spatial and temporal scale clearly does not seem to be a limit, and following your distinction would seem in this case most appropriate?

6) In the parameter description and associated equations (p.4 l.13 to p.5 l.11), you need to be clearer: it is difficult to follow. Directly when you list the eleven parameters I suggest that you use the same order that you use after and that you indicate directly the name of the variables used in the equations (1) and (2). For clarity, these abbreviations have to be everywhere in italics (p. 4 l.22, l.27, 28,...) and called back each time that they are used (e.g. p.5 l.4 for "CO2 concentration within the leaf boundary (Ds)"). Moreover, it is not indicated what the Cs parameter is (equation 2). I suggest also that you indicate how the "stomatal control is affected by soil moisture" (p. 5 l.3).

7) You have to take care about the quality of the figures and tables, and the associated legends (even in the supplementary). The figures have to be clearly understandable. (a) in Figure 1, the WRF grid does not make it possible to see the vegetation around the simulated polygon. I suggest that you indicate in the legend the general location (at least "USA") and the signification of "LS" and "WBS". (b) in table 1 you indicate for the "DBH to Height" an equation with negative "b" value with a negative term "-b x DBH", so the Ht is negative. Moreover you have to give the units of variables (in cm?). (c) in table 3 you use "*" for optimized parameters and for value ranges from EDv2.2. (d) in Figure 3 and 4 you give the number of "days" in "2016". However, it seems not to correspond exactly to a year and it is never explicit: in the text "spring" is for the days
"200 to 250" and in the figure 4 "2016" starts from October (2015?). Please revise the x-axis labelling. (e) in Table S1 you have to indicate clearly the dimensions for each parameter and in a consistent manner (eg "[m]"). (f) in table S2 I suggest that you indicate how the rank is done (by NSE) and that you give the dimension of the parameters. (g) in figure S1 it is not possible to see clearly the differences between simulations. Maybe you could use monthly means?

8) The 2.3 section is called "Inventory and EC tower data" but is mainly about allometric equations. Moreover the approach method to describe shrub allometry can be improved. You suggest that the problem comes from the fact that the model is "originally developed for tropical forest" (p.6 l.9), when it seems to be more precisely due to allometric equations developed for trees and not for shrubs. Then, it could be appropriate to explicitly indicate that from the allometric data available, you transformed them (if I understand well) to a theoretical height considering that the shrub is a cube (?). But more importantly, it could be beneficial if you evaluate the impact of this hypothesis, for example by showing the adequacy between "DBH to Height" results or the height simulated compared to observed height. There is also another solution: to change the allometric equation for shrubs, as is used in other models (e.g. Druel et al., 2017 https://doi.org/10.5194/gmd-10-4693-2017).

9) There is no overlapping between the period of station data (2015-2016) and the years used for the forecast, 2006 to 2014 (p.7 l.12). If it can be understandable to use random years for long term "spinup", using "random years" for all simulations and optimization/validation can introduce a new bias superimposed on the parameter set. Even more important, if you use a random forecast year to simulate specifically 2015 and 2016 (validation and simulation), that means the difference between both simulations is a random year?

10) For the initial parameterisation of the 11 parameters, you choose a sensitivity index. But there are two fundamental hypotheses to use such index: you expect that the responses to the parameters are linear and that there is no interaction between param-

eters. Unfortunately, you never indicate those hypotheses! It is true that at the next step (for optimization) you use a more adapted method (not requiring such hypotheses) and that in the discussion you put two related sentences, but the method is not consistent with the optimization and the hypotheses are required from the beginning. . . The test of the mean of the best sets of 10 parameters shows that the hypotheses were not well considered.

11) You indicate that your "simulations were configured to allow" that other plants than shrubs can grow in the model (p.9 l.1). That means that you specifically activated the competition between species and so other plants can grow? If this is the case, you introduce new uncertainties and so probably directly biases to the optimization and comparison with GPP observations! I really do not understand why you do not use the observed fraction of vegetation in your two (well documented) stations. On the one side you work on very few observations and simulated points (in time and space), but you do not limit the variability induced by the model configuration. Why?

12) The results section suffers from the limitation of the method: only one polygon is simulated, two observation sites considered, with heterogeneous vegetation inside each site (grasses and shrubs) but also between sites (Low Sagebrush /W. Big Sagebrush), and only two years of data (with one not complete for one site), one for simulation and one for validation. Thus, it is not possible to represent inter-annual or spatial variability. Likewise, no statistical tools are used to validate the optimization. We can just observe that one is coherent (WBS) and the other is bad (LS) (the value of the differences are also missing, e.g. p.14 l.5 to 8). In conclusion it is not obvious that the values obtained for the parameters can be used for other years or sites.

13) Not being a specialist of optimization, I cannot say something precisely on this part. But, the choice of the optimization method is not justified or discussed. There exists currently other methods less computationally costly (such as genetic algorithms) and it is possible to extract statistic values to evaluate the efficiency (such as the variability fraction explained before and after the optimization).

14) The discussion allows to go further, but showing mostly the limits of the methods used for the study, which should have been stated earlier in the methods (e.g. the non-linear dependence among parameters). This shows also the gap between the objective indicated (to predict the GPP of sagebrush) and the results (not really validated, even in very restrictive conditions).

Specific comments p.1 l.10. Suggested change: "one of the most critical" to "one critical"

p.1 l.28-31. Suggested change. Remove from "we expect that. . ." (to put in the conclusion?)

p.2 l.3. Need for a reference for "anthropogenic CO2 emissions"

p.2 l.4. Suggested change: Add a small definition for "photosynthesis"

p.2 l.10. Suggested change: "distinct ecosystems" to "distinct ecosystems at large scale"

p.2 l.20. There are currently two spaces after "ecosystems".

p.2 l.27. How do they suppress fire?

p.2 l. 34. After "Great Basin", indicate the density of station (or indicate if there are only two stations. . .)

p.3 l.13. This section (2.1) could gain in clarity if you distinguish (a) the general model presentation (p.3.14 to p.4 l.8) and (b) the presentation of parameters used in this study and their related equation(s).

p.3. l.18 "plant function type" abbreviation is already defined just above (p. 3 l.6).

p.3 l.23. You use acronym "IBMs" which is not defined. Please define it l.21.

p.4 l.13. Suggested change: "parameters. These included" to "parameters:"

p.4. l.18-19. Suggested change: "here we are trying to describe the ones related to

the parameters we have use in this study" to "here describe the ones related to the parameters used in this study"

p.5 l.8. It could be important to state from where the "allometric allocation" comes from, and maybe indicate that they are in Table 1 ?

p.5 l.16. Please clearly indicate where it is (country, state).

p.5 l.16 to 18. If I understood well, you have to indicate that the "200 m x 200 m polygon" is the simulated area in this study (using the 3km resolution WRF forecast). Likewise, in the legend of Figure 1 (p.6 l.2) change "study polygon" to "simulated polygon".

p.6 l.16. Suggested change: Add a line break before the "GPP data. . ."

p.8. l.4. If the sagebrush parameters come only from bibliography, put the citation l.2.

p.8 l.10. Indicate why "370 ppm" or to which year that corresponds (2000?).

p.9 l.29. Change "Fig. 2b and d" by "Fig. 2b, c and d".

p.12 l.21. Change "Table 5" to "Table 6".

p.15 l.4. Suggested change: "was observed" to "was obtained"

p.16 l.16. Suggested change: Add a line break after the "GPP."

p.16 l.20. I am not sure that you can say "quite well".

---

## Author Response (AR1)

**Response to reviewer #1:**

We would like to thank the reviewer for the time to provide a thorough review. We have provided our response for each of the comments (shown in bold) below, to improve the paper.

The manuscript tries to provide a new parameter set for the representation of shrubs in the ED2 – DGVM. The implementation aims to improve GPP estimation in shrublands. Yes, shrublands are under-represent in DGVMs and need more consideration, but I think the present manuscript need an extensive revision to show that shrublands work well within the ED2 model. For two sites a simple methods is used to optimise the parameter values, but the study provide no cross-validation and no further application is given.

This initial study had 2 years of data available from the flux towers (2015-2016), and thus we maximized this available data. Since submittal of the paper, an additional year of flux tower data became available (2017) and we have now included this for subsequent validation in our revision. Additional revisions (please see below) have also been made.

As I have general caveats about the methods used in this study I will list them here and will not go into much detail.

1. Most importantly, the method used here to optimise parameters is not state of the art. There are a lot of methods usually applied to solve the problem of parameter optimization as the Monte Carlo Analysis or genetic optimisation algorithms. Then it would be possible to include all important parameter for the optimisation procedure.

We agree that additional optimizations (and sensitivity) should be performed; for this analysis we used the exhaustive (brute force) method due to computational and study limitations. We spent extensive time on developing the shrub (representing sagebrush) PFT for the EDv2.2 model (e.g. establishing allometric relationships) and several preliminary model run-ups to match with the ecosystem conditions. We've modified the paper to highlight this intent and the conclusions we may draw from the existing work. Again, this research is intended to introduce the sagebrush PFT and its implementation in EDv2.2. Additional robust optimization and sensitivity analyses, and broad spatial scale analysis are the next steps. And we have suggested these in the Discussion and Conclusion sections.

2. Secondly, the same as for the parameter optimisation, the parameter sensitivity measure should be performed with a more comprehensive method (e.g. using partial rank correlation coefficient (PRCC) or Fourier Amplitude Sensitivity Test (FAST)). A freely available paper (https://www.ncbi.nlm.nih.gov/pmc/articles/PMC2570191/) gives a overview of the methods, which can be used to conduct parameter optimisation and sensitivity tests.

We used the Sensitivity Index (SI) which is a straightforward linear (and thus efficient) approach. Again, our intent here was to perform preliminary analyses to demonstrate the sagebrush PFT and more robust analyses will need to take place to demonstrate the value of using ED within sagebrush-steppe. Please also see above responses.

3. Another point is that the authors should use both sites to optimise the parameter set, if they want to apply the model on a broader scale. Furthermore, I didn't understand why the study provides the 10 best ensemble means, these can't be better than the best estimate. But anyhow the authors don't provide a cross-validation. Hence it is impossible to evaluate the performance of the optimised parameters as these are used for the optimisation already.

We agree, and we have optimized the second site (WBS, see section 2.6 and Figure 4) similar to LS and modified the relevant text accordingly. In addition, we calibrated the model using two years of flux tower observation (2015 and 2016) and used 2017 observation for validation, which we did not have at the time of submittal. We agree that we need to perform validation with additional observation sites, in order to evaluate the model performance at the regional scale. However, the two additional observation sites in the region are very different from our calibration sites in terms of vegetation composition and morphology. Our intent here is to present the 10 best simulations for each site to document our results, about the range of parameter combinations and potential reference to further studies on sagebrush PFT.

**Some other important points are striking:**

Metrological data are used for a different time period as the GPP data to which parameters are optimised. If you perform a parameter optimisation specifically for a site, you should use the metrological data of this site, which are normally provided by the EC tower data. But at least the same time period needs to be used.

During submittal, we used random years of the meteorological forcing data (WRF) (from 2005-2015) because the data were not available beyond 2015 for the domain we used (at 3 km). We agree with your comments and, in the revision, used meteorological forcing data (WRF) for the same years as the model simulation years which ranged from 2001 to 2017, using 1 km resolution data.

**The authors state that the equilibrium is reached after 15 years, which seems to be very short. Figure 2 gives a hint that equilibrium is maybe not reached.**

For the previous version of the manuscript, we used eight years for sensitivity analysis which was shown in Figure 2. In this version, we have revised the sensitivity analysis with a 15 year run. A strength of this study is that we are able to initialize the EDv2.2 model using the current state of the ecosystem. In our study we initialized the model with the mean cohort figures based on inventory data (section 2.3. main manuscript) from each of the study sites following approaches similar to other studies (Medvigy and Moorcroft, 2012; Antonarakis et al., 2014). To clarify, we modified the manuscript accordingly (P.8.1.13).

**It is not clear to me if the ED2 model used here includes the nitrogen cycle or if the fire dynamics is turned off for the optimisation procedure.**

We ran the optimization by turning off the fire dynamics in the model. But, it includes the nitrogen and other biogeochemical processes in a DGVM. Please note that in recent years the two sites have not been disturbed by fire.

**It is strongly stated in the introduction that fire dynamics plays an important role in the global carbon balance, but isn't treated in the study!**

Correct, but again this study is focused on developing the shrub PFT and initial ED modeling runs for sagebrush ecosystems. Our study lays the foundation for future studies that can incorporate fire dynamics and other disturbance effects. We have emphasized this in the introduction and conclusions in the revised manuscript.

Authors mentioned that they have changed the allometric equations, but it is never written how, please add that to your manuscript as it is an important information. But also how the used parameter are applied in the model would be a nice additional information. This would help the reader to understand why parameters are sensitive or maybe not.

Thank you for the comment - we provided the shrub allometric equations in P.7.1.7 and additional information here P.6.1.13 to P.7.1.3. We used these coefficients as some of the sagebrush PFT parameters as shown in supplement Table S1.

**Why do you use a different parameter range for optimisation and sensitivity test, or did I get it wrong?**

We used a broader range (based on literature and other land models) in our sensitivity analysis in order to cover the entire range of possible values of the sagebrush parameters. We used these findings to be more efficient and realistic in our optimization. We clarified this in the manuscript section 3.2.

**And how did you define the parameter range? I missed some references here. The TRY database is an extensive source to determine the parameter range.**

We used existing literature for defining sagebrush (or common shrub) parameters, and also a range of parameters for shrub PFTs adopted by other land models (like CLM) to define the parameter ranges (Please see reference column in Table 4). We reviewed the TRY database and they have limited information (eg. sla, shrub height and leaf width) for sagebrush.

**You have not shown any measures in the figures.**

The Bias, RMSE, NSE are in the tables and we have now added the measures (NSE, RMSE) for the best case to the figures, as well.

**And I do not agree that it is a good match for a site-specific optimisation as stated in the manuscript.**

In this revision, we optimized both sites with meteorological data and using representative vegetation conditions from the respective locations trying to match the site conditions (P.8.1.5). Given the complexity of these sites, we feel that the representation of the optimization is sufficient for an initial demonstration.

**Lastly, there are a lot of statements in the abstract and in the introduction about the global importance of shrublands for the global carbon cycle, but authors don't show an application.**

We have revised the manuscript to focus more on the sagebrush PFT development and preliminary performance evaluation of the ED model runs. We have also discussed that with this first step in sagebrush parameterization we could scale up the model performance to regional scales with further refinement in parameterizations (see Conclusions).

**Response to reviewer #2**

We would like to thank the reviewer for the time in doing a thorough review of the manuscript. Our response to each of the comments (shown in bold) is as given below.

**General comments This article is about the optimization of sagebrush parameters based on GPP in the EDv2.2 model and in Great Basin.**

The development and optimization of specific vegetation - here a specific shrub - are currently a key research area to increase model adequacy with observations and enable the simulation of the future development of our ecosystems. However, contrary to that suggested by the actual title, in this article there is no presentation of GPP results estimation but rather only some "optimization and validation" of parameters. Moreover, the article does not present some general case but more a specific situation: a very small zone (200m2 simulated, 2 years of observation in 2 points). I suggest to change the title to make it more explicit. The model used here is EDv2.2, which seems interesting for medium scale simulation. But the methods used limit the scope (and so the interest) of this study. The two differents sites of observation are located close to each other but differ in the type of sagebrush present (a small specie and a big specie). However, only one allometry parameterization is proposed. Your choice to have a dynamic vegetation is curious considering you work on two very specific and well documented sites for one unique year (one for optimization and one for validation). Some of the methods used (as the use of sensitivity index) rely on strong hypotheses, that have been presented only in the discussion. Some deeper bibliography could have made it possible to anticipate errors. The purpose of the article to estimate GPP is thus more a local application of the optimization of parameters in order to simulate (not here) the GPP. Due to the small data set and the validation performed without any statistical test (and one of the two cases that seems not so adequate), there is no insurance that the method could be applied for other years (to predict) or in other sites. As no specific development was presented here, except an adaptation of parameters for sagebrush allometry, the relevance of this article for publication in GMD can be questioned.

We have revised the title of the manuscript to better match the content. The study is primarily focused on the development of shrub (sagebrush) PFT parameters to use in EDv2.2, and to

observe the performance of the model for the newly developed sagebrush PFT (and wherein we used GPP as variable to conduct these comparisons). We agree that allometric relationships for different sites could not properly capture the fine scale heterogeneity in the ecosystem. For this study, we limited our objective in developing general sagebrush parameters, without trying to separate uniqueness of different sagebrush species. We used simple sensitivity and optimization analysis methods, to constrain the selected parameters. In further studies, we intend to capture the non-linear dependencies among these parameters to better constrain them for model estimates; however this is outside the scope of the present study.

**Globally, considering the 14 detailed comments presented below, the editorial and figure quality of the present manuscript, I consider that in this state this article lacks of consistency and does not reach the standard quality expected for GMD.**

Please see our responses below.

**Specific comments**

Not only simulations or field observations can be used to quantify GPP (p.1 l.6). A third essential data set comes from satellites and remote sensing, providing continuous values (spatially and over time). There is for example the GPP from the FLUXCOM project Tramontada et al., 2016 https://doi.org/10.5194/bg- 13-4291-2016 and Jung et al, 2017 https://doi.org/10.1038/nature20780) or from a linear relationship with the Sun-Induced Fluorescence (Su et al., 2017 http://resolver.caltech.edu/CaltechAUTHORS:20171016-145548969). Of course the problem of isolating the GPP for a PFT remains ... as is the case for the observations used in this study. Moreover, this GPP data can be used (if you know the vegetation distribution) to do more efficiency optimization and/or validation (largest temporal and/or spatial scale).

Thank you for the suggestions - we agree that additional data are ideal to quantify GPP. Given the context of this paper (please see comments above), we are limiting our analysis to the flux towers and future work will incorporate the remotely sensed data products and should be useful to assess GPP in broader spatial terms. 2) As indicated in the article (p.2 l.19), it could be difficult in models to represent and parameterize specific ecosystems and they are historically not well simulated. But this is currently a major point of development in land surface models, as for tundra (mosses, shrubs,...) which are now more and more represented. The sentence "Semi-arid, nonforest ecosystems provide an excellent example of this limitation" (p.2 l.20) has to be more documented. More generally, a short review of the current state of what is done in different models would be necessary in this article. Nevertheless, it is probable that these models are not yet sufficient to reproduce specifically the sagebrush.

Thank you, we agree and have additional references cited P.4.1.1.

3) Globally all the references of the article have to be checked. There are wrong dates in the reference list (e.g. for Bradley and Chambers), some references are missing (e.g. Skamarock et al, 2008 and Wright et al., 2004), others are never used in the text (e.g. Brabec et al,2001 and NPS, 2018) and one seems wrong (Davidson et al., 2011 about amazon forest to illustrate tundra). You also have two undifferentiated "USDA, 2018".

Thank you for pointing this out and we have updated the references throughout the manuscript.

4) In the introduction (as suggested in the title) you say that you are going to predict the GPP (p.3 l.4). This seems a little ambitious compared to what is actually done in the result section: an optimisation and validation. In my sense, prediction consists in running the model in the future and simulating the future evolution of GPP.

We have changed the title and agree that we are not predicting GPP but estimating GPP to evaluate the model performance with a sagebrush PFT.

5) At the beginning of the methods (p.3 l.16 to 23), you are doing a distinction between two types of model: "gap" or "big leaf". If the general differentiation between both is clearly understandable, some inaccuracies have to be checked and the references have to be improved / updated. (a) p.3 l.20 and l.22 you indicate that in individual based models you can have competition, coexistence and disturbance, and that it is a limit for the big leaf model. But you have also big leaf models (DGVM) with competition,

disturbance,... (b) p.3 l.23 you indicate that individual-based models have problems due to computing cost, but this is becoming less and less of a problem and currently many large-scale models (initially big leaf) have developed individual based version. Moreover, in this article the small spatial and temporal scale clearly does not seem to be a limit, and following your distinction would seem in this case most appropriate?

Thank you for pointing this out. We agree with the reviewer that there are some "big leaf" models with competition. The challenge with these models, however, is they do not capture the demographic processes such as vertical light competition, competitive exclusion, and successional recovery from disturbance. To make it more clear, we changed the word "competition" in the manuscript to "demographic processes". Considering comments on the IBMs, we agree with the reviewer that computation time is becoming less important in these models. However ED2 is not purely an IBM, as we mentioned in the manuscript (P.3.1.18) its a cohort based model which incorporate different processes.

6) In the parameter description and associated equations (p.4 l.13 to p.5 l.11), you need to be clearer: it is difficult to follow. Directly when you list the eleven parameters I suggest that you use the same order that you use after and that you indicate directly the name of the variables used in the equations (1) and (2). For clarity, these abbreviations have to be everywhere in italics (p. 4 l.22, l.27, 28,...) and called back each time that they are used (e.g. p.5 l.4 for "CO2 concentration within the leaf boundary (Ds)"). Moreover, it is not indicated what the Cs parameter is (equation 2). I suggest also that you indicate how the "stomatal control is affected by soil moisture" (p. 5 l.3).

Thank you for pointing this out. We have added a table (P.4.1.10) to describe parameters we have used for the analysis and put it in a sequential order to match the writing in the test. We also added text to clarify how 'stomatal control is affected by soil moisture' in P.5.1.10. Additionally, we have provided reference (mainly Moorcroft et al., 2001; and Medvigy et al., 2009) for detailed information on equations and processes.

6) You have to take care about the quality of the figures and tables, and the associated legends (even in the supplementary). The figures have to be clearly understandable. (a)

in Figure 1, the WRF grid does not make it possible to see the vegetation around the simulated polygon. I suggest that you indicate in the legend the general location (at least "USA") and the signification of "LS" and "WBS". (b) in table 1 you indicate for the "DBH to Height" an equation with negative "b" value with a negative term "-b x DBH", so the Ht is negative. Moreover you have to give the units of variables (in cm?). (c) in table 3 you use "\*" for optimized parameters and for value ranges from EDv2.2. (d) in Figure 3 and 4 you give the number of "days" in "2016". However, it seems not to correspond exactly to a year and it is never explicit: in the text "spring" is for the days"200 to 250" and in the figure 4 "2016" starts from October (2015?). Please revise the x-axis labelling. (e) in Table S1 you have to indicate clearly the dimensions for each parameter and in a consistent manner (eg "[m]"). (f) in table S2 I suggest that you indicate how the rank is done (by NSE) and that you give the dimension of the parameters. (g) in figure S1 it is not possible to see clearly the differences between simulations. Maybe you could use monthly means?

Thank you for pointing this out. We have updated these figures / tables. (a) we updated figure 1 related to study area which now shows location of LS and WBS sites in Reynold Creek Experimental Watershed (RCEW) area, (b) We removed -ve in the coefficient and provided unit for DBH, (c) in table 4 (earlier 3) we adjusted the confusion with regards to the use of '\*' symbol, (d) we have updated the figures to make it more readable (e) we provided information about NSE score equation used in the ranking (Supplement P.6.1.9). (f) we have provide unit for applicable parameters (Supplement Table S1), (g) we updated the figure (Supplement Fig S2) to show average monthly GPPs to make different simulations more discernible.

7) The 2.3 section is called "Inventory and EC tower data" but is mainly about allometric equations. Moreover the approach method to describe shrub allometry can be improved. You suggest that the problem comes from the fact that the model is "originally developed for tropical forest" (p.6 l.9), when it seems to be more precisely due to allometric equations developed for trees and not for shrubs. Then, it could be appropriate to explicitly indicate that from the allometric data available, you transformed them (if I understand well) to a theoretical height considering that the shrub is a cube (?). But more importantly, it could be beneficial if you evaluate the impact of this hypothesis, for example by showing the adequacy between "DBH to

Height" results or the height simulated compared to observed height. There is also another solution: to change the allometric equation for shrubs, as is used in other models (e.g. Druel et al., 2017 https://doi.org/10.5194/gmd-10-4693-2017).

This is a good idea and we compared the predicted height from the cube root volume with observed sagebrush height using a new set of data from the Great Basin (see Supplement FigS1). We observed a good match between observed and predicted heights for sagebrush.

8) There is no overlapping between the period of station data (2015-2016) and the years used for the forecast, 2006 to 2014 (p.7 l.12). If it can be understandable to use random years for long term "spinup", using "random years" for all simulations and optimization/validation can introduce a new bias superimposed on the parameter set. Even more important, if you use a random forecast year to simulate specifically 2015 and 2016 (validation and simulation), that means the difference between both simulations is a random year?

We used corresponding years of meteorological data for simulation in the revised manuscript. We used 1 km WRF data from 2001 to 2017 for both the sites studied. This will help reduce the interannual uncertainty that may arise from using meteorological data from a random year.

9) For the initial parameterisation of the 11 parameters, you choose a sensitivity index. But there are two fundamental hypotheses to use such index: you expect that the responses to the parameters are linear and that there is no interaction between parameters. Unfortunately, you never indicate those hypotheses! It is true that at the next step (for optimization) you use a more adapted method (not requiring such hypotheses) and that in the discussion you put two related sentences, but the method is not consistent with the optimization and the hypotheses are required from the beginning. The test of the mean of the best sets of 10 parameters shows that the hypotheses were not well considered.

We agree that the chosen method assumes linear dependencies of selected parameters with the target variable. We spent extensive time on developing the shrub (representing sagebrush) PFT for the EDv2.2 model (e.g. establishing allometric relationships) and several preliminary model run-ups to match with the ecosystem conditions. We used the exhaustive (brute force) method due to computational limitations. This study was mainly intended to introduce the sagebrush PFT and its implementation in EDv2.2. We agree that additional robust optimizations (and sensitivity) should be performed. We've modified the paper to highlight this intent and the conclusions we may draw from the existing work. We have added lines to state the limitation of the applied SI method and our assumption on parameters under methods section (P.8.1.18).

10) You indicate that your "simulations were configured to allow" that other plants than shrubs can grow in the model (p.9 l.1). That means that you specifically activated the competition between species and so other plants can grow? If this is the case, you introduce new uncertainties and so probably directly biases to the optimization and comparison with GPP observations! I really do not understand why you do not use the observed fraction of vegetation in your two (well documented) stations. On the one side you work on very few observations and simulated points (in time and space), but you do not limit the variability induced by the model configuration. Why?

The study site is heterogeneous and thus we need to allow additional PFTs to grow to capture total GPP. We do not understand the question here but to clarify we used density information for initialization that has been collected at the sites.

12) The results section suffers from the limitation of the method: only one polygon is simulated, two observation sites considered, with heterogeneous vegetation inside each site (grasses and shrubs) but also between sites (Low Sagebrush /W. Big Sagebrush), and only two years of data (with one not complete for one site), one for simulation and one for validation. Thus, it is not possible to represent inter-annual or spatial variability. Likewise, no statistical tools are used to validate the optimization. We can just observe that one is coherent (WBS) and the other is bad (LS) (the value of the differences are also missing, e.g. p.14 l.5 to 8). In conclusion it is not obvious that the values obtained for the parameters can be used for other years or sites. As stated above, we have simulated both sites with respective ecosystem and atmospheric conditions to address variation between the sites. In our revised analysis, we could use two years of data for calibration and another year for validation. We agree that these are not sufficient to capture inter annual variability but we were mostly limited with the available observation data from the sites. We agree that the values cannot be used for other years and sites until further optimization is performed. We have stated this in the Conclusion (P.17.1.18).

13) Not being a specialist of optimization, I cannot say something precisely on this part. But, the choice of the optimization method is not justified or discussed. There exists currently other methods less computationally costly (such as genetic algorithms) and it is possible to extract statistic values to evaluate the efficiency (such as the variability fraction explained before and after the optimization).

We agree there additional optimization tools could improve the results and provide robust information on sagebrush PFT parameters (Please refer to answers to Q.9 for more).

14) The discussion allows to go further, but showing mostly the limits of the methods used for the study, which should have been stated earlier in the methods (e.g. the nonlinear dependence among parameters). This shows also the gap between the objective indicated (to predict the GPP of sagebrush) and the results (not really validated, even in very restrictive conditions).

Good point, we have tried to clarify the objectives and the results and how our study has contributed to the overall modeling of shrub-steppe (P.2.1.30). We have stated limitation of our tools (P.8.1.18) and potential improvements we would achieve with different methods (P.17.1.2)

**Specific comments**

p.1 l.10. Suggested change: "one of the most critical" to "one critical"

The text has been changed.

**p.1 l.28-31. Suggested change. Remove from "we expect that. . . " (to put in the conclusion?)** As suggested, the lines were removed.

p.2 l.3. Need for a reference for "anthropogenic CO2 emissions"

The text has been removed.

**p.2 l.4. Suggested change: Add a small definition for "photosynthesis"**

We have updated the text.

**p.2 l.10. Suggested change: "distinct ecosystems" to "distinct ecosystems at large scale"**

We changed the texts

**p.2 l.20. There are currently two spaces after "ecosystems".**

We corrected.

**p.2 l.27. How do they suppress fire?**

Removed the text 'suppress fire'

**p.2 l. 34. After "Great Basin", indicate the density of station (or indicate if there are only two stations. . .)**

The text is revised.

**p.3 l.13. This section (2.1) could gain in clarity if you distinguish (a) the general model presentation (p.3.14 to p.4 l.8) and (b) the presentation of parameters used in this study and their related equation(s).**

We tried to differentiate the information in the section through different paragraphs . We added a table showing parameters used in the study followed by brief descriptions and controls of the parameters.

**p.3. l.18 "plant function type" abbreviation is already defined just above (p. 3 l.6).**

We updated the text accordingly.

**p.3 l.23. You use acronym "IBMs" which is not defined. Please define it l.21.**

We corrected as per your suggestion.

**p.4 l.13. Suggested change: parameters. These included" to "parameters:"**

The text has been revised.

**p.4. l.18-19. Suggested change: "here we are trying to describe the ones related to the parameters we have use in this study" to "here describe the ones related to the parameters used in this study"**

We made suggested change.

**p.5 l.8. It could be important to state from where the "allometric allocation" comes from, and maybe indicate that they are in Table 1 ?**

We referred Table 1 for the allometric equation referred in the text P.5.1.18

**p.5 l.16. Please clearly indicate where it is (country, state).**

We updated with region and Country.

p.5 l.16 to 18. If I understood well, you have to indicate that the "200 m x 200 m polygon" is the simulated area in this study (using the 3km resolution WRF forecast). Likewise, in the legend of Figure 1 (p.6 l.2) change "study polygon" to "simulated polygon".

We updated the Figure 1 showing study area.

p.6 l.16. Suggested change: Add a line break before the "GPP data. . ."

We updated with a line break to separate two types of data sources.

**p.8. l.4. If the sagebrush parameters come only from bibliography, put the citation l.2.**

We updated the text to appropriately represent the procedure P.8.1.3. We also updated supplement Table S1 with all PFT parameters to clearly state the source/reference of different parameters.

**p.8 l.10. Indicate why "370 ppm" or to which year that corresponds (2000?).**

we updated the text as suggested (P.8.1.11)

**p.9 l.29. Change "Fig. 2b and d" by "Fig. 2b, c and d".**

We made necessary edits as suggested.

**p.12 l.21. Change "Table 5" to "Table 6".**

We made necessary corrections.

**p.15 l.4. Suggested change: "was observed" to "was obtained"**

Changed the text P.16.1.10.

**p.16 l.16. Suggested change: Add a line break after the "GPP."**

We made the edits as suggested.

**p.16 l.20. I am not sure that you can say "quite well".**

Text has been updated.

**Developing and optimizing Optimizing shrub parameters representing to estimate gross primary production of the sagebrush (*Artemisia* spp.) ecosystems in the Northern Great Basinecosystem using the Ecosystem Demography (EDv2.2) model**

5 Karun Pandit1, Hamid Dashti1, Nancy F. Glenn1, Alejandro N. Flores1, Kaitlin C. Maguire2, Douglas J. Shinneman2, Gerald N. Flerchinger3, Aaron W. Fellows3

1Department of Geosciences, Boise State University, 1910 University Dr., Boise, ID 83725-1535 USA
 2United States Geological Survey, Forest and Rangeland Ecosystem Science Center, 970 Lusk St., Boise, ID 83706
 3United States Department of Agriculture, Agricultural Research Service, 800 Park Blvd., Suite 105, Boise, ID 83712

**10 Correspondence to: Karun Pandit (karunpandit@gmail.com)**

[revised manuscript text omitted]

**Detailed**

As we can find detailed descriptions of sub-models of EDv2.2 are available in the existing literature (Medvigy et al., 2009; Moorcroft et al., 2001); thus,); here we are trying to describe the ones related to the parameters we have used in this study. The ecophysiological sub-model has a coupled photosynthesis and stomatal conductance scheme developed by Farquhar and Sharkey (1982) and Leuning (1995).) respectively, and which estimates takes care of leaf-level carbon and water fluxes. Leaf-level carbon demand of C3 plants is determined by the minimum of light-limited rate (Je) and Rubisco-limited rate (Jc), and  $V_{m0}$  controls the later as given by Eq.(1) after being scaled to a given temperature.

$$J_{c} = \frac{V_{m}(T_{v})(C_{inter} - \Gamma)}{C_{inter} + K_{1}(1 + K_{2})}$$
(1)

10

15

20

5

where,  $V_m(T_v)$  is the maximum capacity of Rubisco to perform carboxylase function at a given temperature  $T_v$  scaled from  $V_{m0}$ (Medvigy et al., 2009)1,7  $C_{inter}$  is the intercellular CO2 concentration,  $\Gamma$  is the compensation point for gross photosynthesis, K1 is the Michaelis-Menten coefficient for CO2, and K2 is proportional to the Michaelis-Menten coefficient for O2. Stomatal conductance which is modeled using Leuning (1995), a variant of Ball Berry model (Eq. 2), is influenced by *stomatal slope* and *cuticular conductance*.

$$g_{sw} = \frac{MA_o}{(C_s - \Gamma) (1 + \frac{D_s}{D_0})} + b$$
(2)

where,  $A_o$  is photosynthetic rate,  $g_{sw}$  is stomatal conductance for water,  $A_o$  is photosynthetic rate, M is stomatal slope, b is cuticular conductance,  $D_0$  is empirical constant,  $D_s$  is water vapour deficit and CO2 concentration within leaf boundary, and  $\Gamma$  is as described above. Stomatal control is also affected by soil moisture supply term, which is a function of soil moisture, fine root biomass, and water conductance. When the available water supply is less than the demand predicted by photosynthesis-conductance model, then photosynthesis, transpiration, and stomatal conductance are all linearly weighted down to match the supply (Dietze et al., 2014).

Water and CO2 concentrations within the leaf boundary layer areis influenced by *leaf width* along with other factors like wind speed, leaf area index, and molecular diffusivity of heat. *Specific leaf area* (SLA) has units of leaf areaare per unit leaf carbon and is used to scale up leaf-level <del>fluxes</del> to canopy-level fluxes. Relationships Relationship between growth respiration and net photosynthesis areis controlled by the *growth respiration factor*. In EDv2.2, while leaf biomass is determined by PFT specific allometric equation (as shown in Table 2 for sagebrush)equations based on diameter, fine root biomass is defined by a ratio of leaves to fine roots. Leaf turnover and fine root turnover rates together influence overall litter turnover rate, even though in deciduous trees dropping of leaves also affects this rate. Turnover rate of stored leaf pool and storage respiration depends on storage turnover rate, size of stored leaf pool, and storage biomass.

**5**

10

15

**2.2 Study area**

While the PFT was developed broadly We initialized and performed parameter optimization for sagebrush, we focused ecosystems in the EDv2.2 model runs at the RCEW site, using field data and two EC station sites located in the Northern Great Basin region of Western United States (Fig. 1). This site is Reynolds Creek Experimental Watershed (RCEW) and Critical Zone Observatory (CZO), operated by the USDA Agricultural Research Service and is also a Critical Zone Observatory (CZO) (referred to as RC-CZO). (Fig. 1). We used twoa 200 m x 200 m polygons centered at two EC sites within RC-CZO to closely represent the footprint area of these sites. polygon with center location of 43.15 N and 116.72 W and a mean elevation of 1583 m. The AmeriFlux US-Rls EC station, located at 43.1439 N and 116.7356 W and at an elevation of 1583 m, is within the Lower Sheep Creek drainage in RCEW.is approximately 0.7 km from the center of our study site. The area within the footprint of this sitetower is dominated by low sagebrush (Artemisia arbuscula) and Sandberg bluegrass (Poa secunda) (Stephenson, 1970; Seyfried et al., 2000) and is characterized as having light cattle grazing (AmeriFlux, 2018). The second Another AmeriFlux tower, US-Rws, is located at 43.1675 N and 116.7132 W in the Nancy Gulch drainage, within about 2.5 km distance

to the northeast of from the US-Rls sitestudy polygon. This area is dominated by Wyoming big sagebrush (A. tridentata ssp. wyomingensis) and bluebunch wheatgrass (Pseudoroegneria spicata) (Stephenson, 1970). Hereafter, these two sites are 20 designated as LS (for low sagebrush) and WBS (for Wyoming big sagebrush), respectively.

---

## Referee Report (RR1)

Review of:
Developing and optimizing shrub parameters representing sagebrush (Artemisia spp.) ecosystems in the Northern Great Basin using the Ecosystem Demography (EDv2.2) model
By:Karun Pandit et al.

**General comments**

This article is about the development and the parameterization (through the optimization) of shrubs in the EDv2.2 model. It focuses on two sites in the Northern Great Basin and on the sagebrush. The introduction of new vegetation descriptions in models is a key step to improve and simulate ecosystems more precisely. This article presents at least one first step of the introduction of shrubs in EDv2.2, necessary to represent some ecosystems (present in Great Basin but also throughout the world).

There is an undeniable improvement compared to the first version submitted a few months ago, both in form and content. The title is in accordance with the article, the figures are clearly more readable, more coherent tests were provided and discussion/conclusions are more relevant. However, it seems to me that this could have been the first version submitted. And despite the deep work provided in this version, some comments were quickly replied to and sometimes without real justification (e.g. for the references).

Nevertheless, the paper can still be improved in precision and quality, by adding some coherence to the study with a (relatively) small amount of work. I listed below my comments.

**Comments**

1) There is an important improvement with the use of more yearly data forcing, removing some biases due to random year forcing used between optimisation simulations. The discussion was also clearly improved, taking into account the methodological limits. Nevertheless, the choices of the methods used still have to be better justified in some case. Indeed, as it stands, it is not obvious that all tools are well considered. The main point is the justification of the use of "brute force" for the optimization due to computing limitation (p.9 l.11), while other methods are available especially in order not to try every configuration of the parameter set but to chose a new set of parameters depending the previous sets tested. But for this purpose some assumptions with the variables (mostly the independence between parameters) have to be made (like for your sensitivity test). You can therefore justify you method and discuss about other possibilities and your hypothesis for the sensitivity test in the discussion. Another point is the choice of using an ensemble mean parameter and best cases. Why have you done this choice? For/with which hypotheses? Also in the discussion it could be interesting to compare and discuss the two sets of parameters (differences, results…).

2) As proposed before, it seems to me that the discussion could go a little further. For instance, you could compare the two sites and the two sets of parameters to introduce the following

idea: which set of parameters will you use if you need to use your model in another situation or to start another optimisation? It could be interesting also to discuss about the choice to change only the parameters and allometry to discriminate shrubs from trees. Can the important differences with the observations shown be due to a missing process?

3) It was a good idea to introduce a table (Table 1.) to present the parameters used in the equations, for the sensitivity test and the optimization. The text would be more readable if you add inside the table the abbreviations used in the text/equations. Moreover, there is still inconsistency in the use of italics for parameter abbreviations (some in italics and other without) and it is still not indicated what the Cs parameter is (equation 2). Note that in the table for the units the exponents disappeared and it could be important to indicate what the "a" unit stands for.

4) In this new version, you have changed the number of steps for $V_{m0}$ (Table 5.) and so the number of simulations (p.12 l.13) but with the same justification as in the previous version. How is it possible? In regards to the substantial changes in the simulations, it is not surprising to change the configurations but the scientific approach can be questioned. This new choice has to be indicated (at least in the authors response).

**Specific comments**

p.2 l.5. Need for a reference for this sentence.

p.2. l.28. It is important to indicate that Great Basin is in the USA.

p.3 l.13-21. Now better introduced. Do not hesitate to explain why it could be interesting to work with cohort (to insist about the advantages of the ED2 model type).

p.5 Figure 1. It is difficult to read the name of the EC towers inside the figure.

p.13 l.2-3. To be consistent, the SD values have to be indicated in Table S2.

p.13 l.7-8. It could be interesting to compare both cases, and so the simulation with the mean parameters can be added in Figure 3 (which was present in the previous version).

p.15 l.6. Suggested change: remove 'also'.

p.16 l.27. There is no Figure 5. (Fig 3 and 4?).

Fig. S2. The figure is a lot more readable than before. However, it could be probably better if you highlight the best parameters. Another possibility is to show instead of the other 9 curves the mean of the 10 curves with the SD (+ curve of the ensemble parameters).

Fig. S3. It could be very interesting (to support your sentence p.13 l.6) to add in the same type of graph the fraction of shrubs and grasses.

---

## Author Response (AR2)

**Reviewer #1,**

We would like to extend our thanks to the reviewer for taking time to review the revised manuscript. We have tried to provide our response to the comments (shown in bold) below, to improve the paper.

**Suggestions for revision**
**The authors conscientiously responded to the referees' comments, especially that they have strengthened the focus of the manuscript on the implementation of the shrub PFT and the more local application.**

We would like to thank the reviewer for this comment.

**However, I still do not understand why the brute force method is used because it is the most computational expensive method, particularly as the authors argue that due to computational limitation further optimization is not feasible for this study.**

We agree that brute-force approach is computational expensive. We tried to reduce the candidate solutions to bring the computational cost to a manageable size by exploring only five most sensitive parameters and by limiting their ranges. Please see a more detailed response below (Answer to Reviewer 2) regarding some of the alternative methods and our justification for using the brute-force method. Also note that we have added more discussion of this approach in the methods (P.9.l.26-P.10.l.6), and discussion (P.20.l.19-23).

**For the broader scale, I miss a suggestion which parameter should be used and how this would affect the different sites. This would at least ensure that the new sagebrush PFT can be used at the regional scale.**

We identified ten best parameter combinations that estimated GPP closest to the observation for each of the two EC sites. With regards to recommending appropriate parameters for regional analysis, we think we still need to cover more study sites and adopt advanced sensitivity and optimization tests before we can recommend the best parameters for large scale regional analysis (Please refer to Conclusions in the manuscript). However, if we have prior (supporting) information about the dominant sagebrush type in a particular area of interest, we can choose the most suitable parameters from among our list of identified parameter combinations for parameterization (P.21.l.5-7).

**Only for a better understanding: The ten best simulations seem to change fairly linearly and differ not much (Figure S2). The parameter used for the optimization does not appear to change the dynamics of the GPP that might be due to the limited choice of parameters and the measure to rank the parameter used for the optimization.**

We selected eleven different parameters closely related to GPP based on several previous studies dealing with parameterization of process-based models. We agree that we may have lost

considerable amount of information when we selected the top five most sensitive parameters and excluded others. We have touched upon this limitation and as a potential area of further improvement in the discussion section (P.19.l.18).

Process-based models like EDv2.2 are ill-posed where we can obtain almost the same results with different sets of model parameterizations (P.10.l.24-29), which may be the reason we did not see much variation among the top ten ensemble members in Figure S2. Also, the results in Figure S2 are only for the top ten simulations and do not completely reveal the effect of parameters throughout the ranges of investigation. As discussed in the manuscript, our current study was focused mostly on developing the sagebrush PFT parameters, and preliminary sensitivity and optimization tools for the analysis.

**Minor comment: Figure S1 - To which site does this data belong to?**

The data in this graph (Figure S1) is from Sierra Nevada Mountains, California in the Great Basin (Qi et al., 2018) which represents the southern extent of sagebrush communities in the region (p.7.l10-12).

**Reviewer # 2:**

We would like to thank the reviewer for the time to provide a thorough review on the revised manuscript. We have tried to provide our response for each of the comments (shown in bold) below, to improve the paper.

**Review of: Developing and optimizing shrub parameters representing sagebrush (Artemisia spp.) ecosystems in the Northern Great Basin using the Ecosystem Demography (EDv2.2) model By:Karun Pandit et al.**

**General comments This article is about the development and the parameterization (through the optimization) of shrubs in the EDv2.2 model. It focuses on two sites in the Northern Great Basin and on the sagebrush. The introduction of new vegetation descriptions in models is a key step to improve and simulate ecosystems more precisely. This article presents at least one first step of the introduction of shrubs in EDv2.2, necessary to represent some ecosystems (present in Great Basin but also throughout the world).**

**There is an undeniable improvement compared to the first version submitted a few months ago, both in form and content. The title is in accordance with the article, the figures are clearly more readable, more coherent tests were provided and discussion/conclusions are more relevant. However, it seems to me that this could have been the first version submitted. And despite the deep work provided in this version, some comments were quickly replied to and sometimes without real justification (e.g. for the references).**

We would like to apologize for our lack of attention to previous comments and thank you for the more favorable assessment of this version of manuscript.

**Nevertheless, the paper can still be improved in precision and quality, by adding some coherence to the study with a (relatively) small amount of work. I listed below my comments.**

**Comments**

**There is an important improvement with the use of more yearly data forcing, removing some biases due to random year forcing used between optimisation simulations. The discussion was also clearly improved, taking into account the methodological limits.**

Thanks!

**Nevertheless, the choices of the methods used still have to be better justified in some case. Indeed, as it stands, it is not obvious that all tools are well considered. The main point is the justification of the use of "brute force" for the optimization due to computing limitation (p.9 l.11), while other methods are available especially in order not to try every configuration of the parameter set but to chose a new set of parameters depending the**

**previous sets tested. But for this purpose some assumptions with the variables (mostly the independence between parameters) have to be made (like for your sensitivity test). You can therefore justify you method and discuss about other possibilities and your hypothesis for the sensitivity test in the discussion.**

The brute-force method was chosen because while it has limitations, it also provides robust estimates of global optimums. The brute-force method tries to explore all possible combinations within pre-defined parameter domains, allowing outcomes for all types of interactions among uniformly distributed parameters. This method can also be computationally expensive, however, if we can limit our search range and run through coarser search steps, the computational time is reduced significantly. This method can provide better estimates of global optimums with less uncertainties compared to emulators or other sample-based methods like Genetic Algorithms.

Alternatives to the brute-force approach include Bayesian methods, which are mostly based on Markov Chain Monte Carlo (MCMC) and require $10^4$ to $10^7$ model runs. These analyses demand a huge amount of computational costs (Dietze et al., 2018; Fer et al., 2018). Emulator methods (surrogate methods) use statistical approaches to build relationships between input parameters and output variables, and reduce the time of running complex models. Similar to the brute-force method, these approaches also have limitations. For example, they may fail to converge in situations where the relationship between parameters and model outputs is highly nonlinear.

We have added these statements in the manuscript in the methods section (P.9.l.26-P.10.l.6), and added text in discussion (P.20.l.19-23) to state the limitations in the method adopted.

**Another point is the choice of using an ensemble mean parameter and best cases. Why have you done this choice? For/with which hypotheses? Also in the discussion it could be interesting to compare and discuss the two sets of parameters (differences, results…).**

With the ensemble parameters, basically, we were trying to capture the non-linearity and high variability in parameter combinations observed from the models runs. Since the process-based models like EDv2.2 are ill-posed they do not necessarily have a unique solution for parameters, so we tried to explore multiple possible values close to the best ones. We updated the text accordingly (please refer p.10.l.24-28). We also modified the meaning for the ensemble mean in this version (p.13.l.19-p.14.l.2). Earlier, we presented a new simulation that used mean parameter combinations, but this approach is likely not applicable for non-linear parameters. In addition, it did not produce better results when compared to the best case. In this new version of interpretation, we compared the best case scenario with ensemble mean (mean estimates from ten best simulations), where we found the results more comparable to the best case. In fact, for the calibration, the ensemble mean performed similar to best cases (even marginally better) in both study sites. We also stated standard deviation associated with each of the mean ensemble parameters (Table 6) to understand the variation.

We updated Figure 3 and Figure 4 to include the mean of ten best simulations, one standard deviation, and range of minimum and maximum, along with best case simulation and observation. This update is also consistent to the comment below regarding Figure S2. We also added monthly mean comparisons for calibration (Figure 3) to make it consistent with the validation figures (Figure 4).

In connection to the application of the parameters, if we were to run EDv2.2 for a certain area, where we have better knowledge on some parameter estimates (eg. Vm0, or SLA) we can choose those (from best ten sets) that are most suited with the local conditions (please refer p.21.l.5-7).

From Table 7 it is obvious that both the best solution and the ensemble of the 10 best solutions leads to almost similar results in terms of errors. However, the estimated parameters for each approach is slightly different (Table 6). Choosing the 10 best solutions gave us higher probability of getting close to global optimization.

Per the comment, we have added more discussion regarding the comparison of different sets of parameters in the Discussion (p.19.l.7-14).

**2) As proposed before, it seems to me that the discussion could go a little further. For instance, you could compare the two sites and the two sets of parameters to introduce the following idea: which set of parameters will you use if you need to use your model in another situation or to start another optimisation? It could be interesting also to discuss about the choice to change only the parameters and allometry to discriminate shrubs from trees. Can the important differences with the observations shown be due to a missing process?**

We added text in the discussion section to compare and interpret two sets of parameters (ie. Best-case and ensemble mean case), within and among two different study sites (p.19.l.7-14). We also tried to suggest potential parameters that can be used to initialize other optimization studies.

We have stated our assumptions while simulating shrubs with EDv2.2 (please refer to p.4.l.7-10) and also discussed the shortcomings of our assumptions and possible future work to improve our results (please see p.19.l.19-25).

**3) It was a good idea to introduce a table (Table 1.) to present the parameters used in the equations, for the sensitivity test and the optimization. The text would be more readable if you add inside the table the abbreviations used in the text/equations. Moreover, there is still inconsistency in the use of italics for parameter abbreviations (some in italics and other without) and it is still not indicated what the Cs parameter is (equation 2). Note that in the table for the units the exponents disappeared and it could be important to indicate what the "a" unit stands for.**

Thank you for noting these inconsistencies. As per the comment, we have included abbreviations used in the texts/equations in Table 1. We also added equation 2 which elaborates how $V_{m0}$ is

related to $V_m(T_v)$ in equation 1. We have also added missing definition of $C_s$ in the equation 3 (previously equation 2), and clearly stated in the Table 1 that $a^{-1}$ stands for per annum.

**4) In this new version, you have changed the number of steps for Vm0 (Table 5.) and so the number of simulations (p.12 l.13) but with the same justification as in the previous version. How is it possible? In regards to the substantial changes in the simulations, it is not surprising to change the configurations but the scientific approach can be questioned. This new choice has to be indicated (at least in the authors response).**

Thank you for the comment. In this revised version, we decreased the upper range of $V_{m0}$ to 14 instead of 11.5 which we adopted earlier, thus reducing the total number of steps from 5 to 4. This was done to lower our computational burden based on our previous optimization analysis where we did not find simulations with 11.5 coming closer to the observed data we used for calibration. Since, we performed simulations (optimization) for both EC sites, by excluding a total of 180 simulations from each of the EC site stations significantly reduced our computational time. However, we did not make any changes to our approach or method of optimization.

**Specific comments**

**p.2 l.5. Need for a reference for this sentence.**

We have included reference for the sentence (p.2.l.4-6).

**p.2. l.28. It is important to indicate that Great Basin is in the USA.**

We added "of the United States" in the sentence to indicate Great Basin in in USA (p.2. l.28).

**p.3 l.13-21. Now better introduced. Do not hesitate to explain why it could be interesting to work with cohort (to insist about the advantages of the ED2 model type).**

We added text to highlight use of cohort based model like EDv2.2 (p.3.l.22-24)

**p.5 Figure 1. It is difficult to read the name of the EC towers inside the figure.**

We have updated the Figure 1 by increasing the size of fonts for the names of EC towers.

**p.13 l.2-3. To be consistent, the SD values have to be indicated in Table S2.**

We have included SD values along with other indicators in Table S2. We also added goodness-of-fit results for validation data for both the study sites in Table S2.

**p.13 l.7-8. It could be interesting to compare both cases, and so the simulation with the mean parameters can be added in Figure 3 (which was present in the previous version).**

We included mean and ±SD for the 10 best (ensemble) simulations along with the best-case scenario. We also updated the comparisons between best-case and ensemble mean in the text.

**p.15 l.6. Suggested change: remove 'also'.**

We removed 'also' from the text.

**p.16 l.27. There is no Figure 5. (Fig 3 and 4?).**

We corrected it to "(Fig 3 and Fig 4)" p. 19.l.33.

**Fig. S2. The figure is a lot more readable than before. However, it could be probably better if you highlight the best parameters. Another possibility is to show instead of the other 9 curves the mean of the 10 curves with the SD (+ curve of the ensemble parameters).**

We changed the figure to show simulation with best parameters, mean of 10 simulations, and ±SD from the mean.

**Fig. S3. It could be very interesting (to support your sentence p.13 l.6) to add in the same type of graph the fraction of shrubs and grasses.**

We agree that it would be interesting to produce a graph showing standard deviation of mean separately for shrub and grass fractions. However, we observed grasses in only a couple of simulations (including best case) in very limited amount compared to shrubs. There would not be much information to add with rest of the nine simulations in terms of mean and variation, so we suggest keeping the current version.

[revised manuscript text omitted]
.2 m̶o̶d̶e̶l̶ u̶s̶i̶n̶g̶ t̶h̶e̶ e̶n̶s̶e̶m̶b̶l̶e̶ m̶e̶a̶n̶ are often ill-posed, meaning that there may not be a unique solution of parameter v̶a̶l̶u̶e̶s̶ a̶n̶d̶combinations but rather several combinations of parameters produce the same solution. One way to solve the ill-posed problem is by selecting more than one of the best c̶a̶s̶e̶ (̶h̶i̶g̶h̶e̶s̶t̶ N̶S̶E̶)̶ p̶a̶r̶a̶m̶e̶t̶e̶r̶ v̶a̶l̶u̶e̶s̶ f̶o̶r̶ b̶o̶t̶h̶ E̶C̶ s̶i̶t̶e̶s̶.combinations, from which we can either explore average outputs or select one of the ensemble members that would better match any prior information such as any correlation among parameters, available data, vegetation characteristics or ecosystem conditions (Combal et al., 2002; Quan et al., 2015). The simulated GPP from these runs were then compared against respective EC site data from 2017, which was withheld from the optimization as a means of providing an independent validation.

**3 Results**

**3.1 Initial parameterization and sensitivity analysis**

[revised manuscript text omitted]

We selected ten simulations with the best NSE scores for both the LS and WBS sites (Table S2 and Fig. S1 in the Supplement) and determined ensemble means of parameter values for these sites (Table 6). To perform validation of these ten best simulations from each EC station, we extended the model runs to obtain GPP estimates for the year 2017. We then ran EDv2.2 from 2001 to 2017 using parameterscompared the biases and skill scores associated with the highest NSE scoretop performing simulation (hereafter, the 'best case') and ensemblethe mean parametersfrom all ten simulations (hereafter, the 'ensemble case') for each of the EC stations.
[revised manuscript text omitted]